# Outflow of cerebrospinal fluid is predominantly through lymphatic vessels and is reduced in aged mice

Qiaoli Ma[1], Benjamin V. Ineichen[2], Michael Detmar [1] & Steven T. Proulx [1]

Cerebrospinal fluid (CSF) has been commonly accepted to drain through arachnoid projections from the subarachnoid space to the dural venous sinuses. However, a lymphatic component to CSF outflow has long been known. Here, we utilize lymphatic-reporter mice and high-resolution stereomicroscopy to characterize the anatomical routes and dynamics of outflow of CSF. After infusion into a lateral ventricle, tracers spread into the paravascular spaces of the pia mater and cortex of the brain. Tracers also rapidly reach lymph nodes using perineural routes through foramina in the skull. Using noninvasive imaging techniques that can quantify the transport of tracers to the blood and lymph nodes, we find that lymphatic vessels are the major outflow pathway for both large and small molecular tracers in mice. A significant decline in CSF lymphatic outflow is found in aged compared to young mice, suggesting that the lymphatic system may represent a target for age-associated neurological conditions.

[1] Institute of Pharmaceutical Sciences, Swiss Federal Institute of Technology, ETH Zurich, 8093 Zurich, Switzerland. [2] Brain Research Institute, University of Zurich, 8057 Zurich, Switzerland. Correspondence and requests for materials should be addressed to S.T.P. (email: steven.proulx@pharma.ethz.ch)

Cerebrospinal fluid (CSF) is considered to be produced primarily by the choroid plexus and to flow through the ventricular system to reach the subarachnoid space (SAS) of the skull and spinal column[1]. An additional source of CSF is the interstitial fluid (ISF) from the brain tissue that is produced at the blood–brain barrier and may account for around 10% of the total volume of CSF[2]. Three layers of meninges surround the central nervous system (CNS), the pia mater that covers the brain and spinal cord parenchyma, and the arachnoid and dura mater layers, which line the skull and vertebral canal. The CSF flows through the SAS between the pia mater, which is semi-permeable, and the arachnoid, which forms a tight barrier to prevent flow into the dura mater[3,4]. Within the CNS itself, lymphatic vessels are believed to be absent, instead a system of paravascular spaces (historically known as Virchow–Robin spaces but recently described as part of the "glymphatic system") is thought to provide channels for the movement of fluid and solutes from the brain interstitial space and to the CSF and vice versa[5].

Based upon work in the early part of the twentieth century, the predominant outflow of CSF from the SAS is accepted to take place through arachnoid villi or granulations that project into the dural venous sinuses[1,6,7]. These structures are described as a series of tubules within an outgrowth of arachnoid tissue that are continuous with the CSF of the SAS of the cranium and the spine[8,9]. They were initially believed to act as one-way valves for a pressure-driven flow from CSF to venous blood[10]; however, evidence from electron microscopy later indicated the presence of an intact barrier of endothelial cells[8,11]. Extensive research has not led to a consensus on the exact mechanism for flow through arachnoid villi and granulations and, despite their widespread acceptance as major outflow sites of CSF, direct in vivo physiological evidence of their function is lacking[12–14].

Since an original report by Schwalbe in 1869, a large body of work in many different species has indicated a role for lymphatic vessels draining CSF in both cranial and spinal regions[15–17]. Attempts to quantify the proportion of CSF drained by the lymphatic system using cannulation of deep cervical lymphatic vessels and radiolabeled tracers have indicated that, in some species such as rabbit and sheep, lymphatic vessels were responsible for around 30–50% of total outflow, with the remainder assumed to be drained through arachnoid villi[18,19]. The outflow pathways were defined using tracers to be within sheaths around cranial nerves, with the cribriform plate route along olfactory nerves that extends to lymphatic vessels in the nasal mucosa deemed especially important[17,20,21]. These sheaths enclose extensions of the SAS that project extracranially with the nerves through foramina of the skull. At these locations, it was suggested that there are pathways for tracers to penetrate the arachnoid membrane to reach interstitial spaces[20,22–25] or possibly directly to lymphatic vessels outside the skull[21,26,27]. Additional outflow pathways were found to be present around spinal nerve roots to reach lymphatic vessels in epidural tissue[28,29]. Two recent reports in mice have indicated that the dura mater of the CNS is endowed with a network of lymphatic vessels capable of draining CSF or brain ISF[30,31]. However, this potential route for drainage of CSF is controversial in light of the existence of the arachnoid barrier layer between the SAS and the dura mater[4].

Thus, the current paradigm suggests a dual-outflow system for CSF to reach the blood circulation, one directly to venous blood through arachnoid projections and one indirectly through the lymphatic system[7]. At this point, however, there is a lack of consensus on what the relative contribution for each of these outflow pathways is to total outflow of CSF. It is also not clear which pathways to reach the lymphatic system are most important. These are key questions as a functioning lymphatic vascular system draining CSF and brain ISF could be essential for many aspects of neurology including immune function and clearance of toxic proteins such as amyloid beta. As many neurological conditions are associated with the aging process, it is also critical to determine if lymphatic outflow of CSF is altered in aged conditions.

We have recently developed near-infrared (NIR) tracers that in conjunction with lymphatic vessel reporter mice, allow high-resolution imaging and quantification of the anatomy and function of lymphatic vessels[32,33]. These bright pegylated NIR dyes avoid many of the complications of other tracers, such as direct blood vessel uptake, adherence to tissue components, or phagocytosis by macrophages, allowing quantification of lymphatic flow

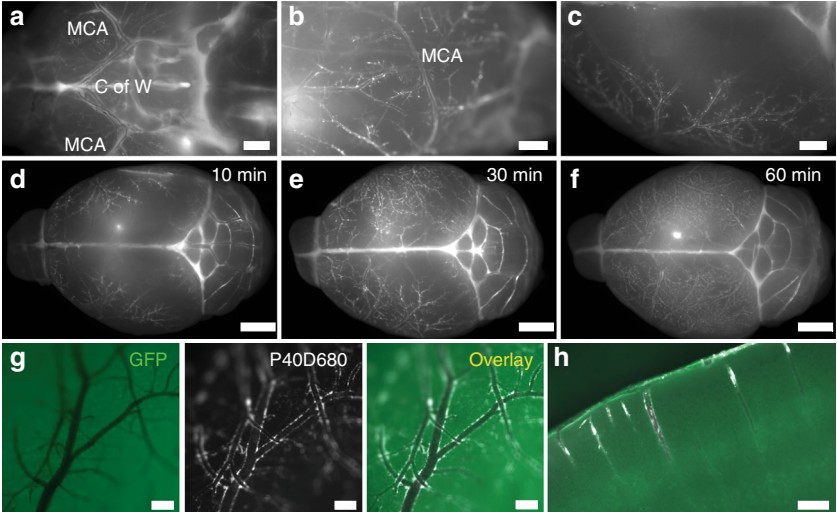

**Fig. 1** Paravascular localization of P40D680 on the pial surface and in the cortex. **a–c** Images of paravascular P40D680 on the ventral, lateral, and dorsal surfaces of the brain. Mice were euthanized 10 min after the completion of intraventricular infusion of P40D680. Images are representative of $n = 5$ mice. Scale bars: 1 mm. MCA: middle cerebral artery. C of W: Circle of Willis. **d–f** Images of paravascular P40D680 at $t = 10$ min, 30 min, and 60 min after infusion. Representative of $n = 5$ mice of each group. Scale bars: 2 mm. **g** Representative detailed image of the dorsal surface of the brain showing paravascular localization of P40D680 at $t = 60$ min. GFP label represents autofluorescence channel. Scale bars: 200 μm. **h** Representative GFP and P40D680 overlay image of penetrating arteries and paravascular localization of P40D680 at $t = 60$ min. Scale bar: 200 μm

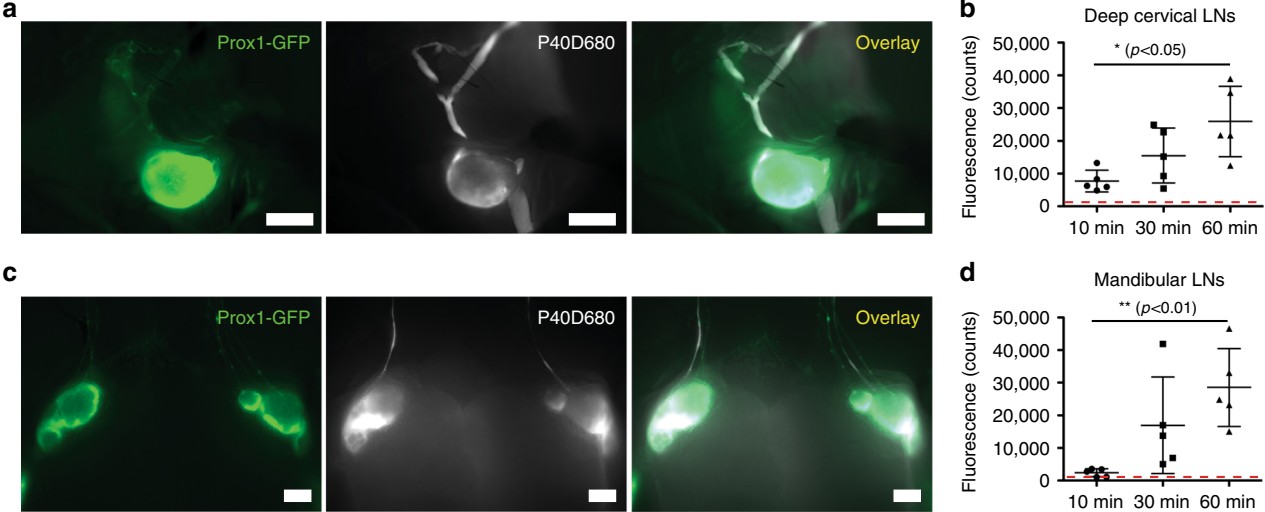

**Fig. 2** Tracer outflow to draining lymph nodes in the neck. **a** Representative pictures of tracer within the left deep cervical lymph node and afferent and efferent lymphatic vessels at euthanization at 10 min after P40D680 infusion into the right lateral ventricle of a Prox1-GFP mouse. Scale bars: 1 mm. **b** Quantification of fluorescent signal of deep cervical lymph nodes at $t = 10$ min, 30 min, and 60 min after infusion. $n = 5$ mice each time point. Red dashed line indicates baseline values from uninjected mice. Data are the mean ± SD. *$p < 0.05$ (one-way ANOVA with the Tukey's multiple comparison). **c** Representative pictures of tracer within the mandibular lymph nodes and afferent and efferent lymphatic vessels at euthanization at 30 min after P40D680 infusion into the right lateral ventricle of a Prox1-GFP mouse. Scale bar: 1 mm. **d** Quantification of fluorescent signal of mandibular lymph nodes at $t = 10$ min, 30 min, and 60 min after infusion. $n = 5$ mice each time point. Red dashed line indicates baseline values from uninjected mice. Data are the mean ± SD. **$p < 0.01$ (one-way ANOVA with the Tukey's multiple comparison)

from the interstitial tissue of many different organs[34,35]. In this study, we first aim to use these tracers to define the major flow routes of CSF after infusion into a lateral ventricle in mice. Next, we use a recently developed tracer transport to blood assay to evaluate the dynamics of CSF outflow to the systemic circulation in comparison to visualization of outflow to collecting veins or lymphatic vessels in an attempt to shed light on the relative contribution of each pathway to total CSF outflow.

We find that the major outflow pathway from CSF for both large and small molecular tracers in mice is through perineural routes to reach the lymphatic system. Furthermore, we demonstrate significantly slower dynamics of CSF outflow in aged 18-month-old mice compared to young 2-month-old controls. Our findings implicate that the lymphatic system has a greater than previously acknowledged role in many neurological conditions.

## Results

**Characterization of CSF flow routes within the cranium.** As CSF is considered to be produced primarily by the choroid plexuses that line the ventricular system, we aimed to detail the flow of the tracer after intraventricular infusions of 40 kDa sized poly(ethylene) glycol IRDye680 (P40D680), a pegylated NIR tracer[32]. Using a stereotactic injector, 2.5 µL of 200 µM P40D680 was delivered to the right lateral ventricle over the course of 2.5 min in Prox1-GFP mice, a reporter line for lymphatic vasculature (Supplementary Fig. 1a–c). A quality check of the ventricle infusion was performed after completion of each experiment by fluorescence imaging of serial 100 µm sections of the brain under a stereomicroscope (Supplementary Fig. 1d–g). A successful ventricular infusion showed a remnant of tracer from the needle penetration through the cortex with no tracer remaining within or lining the ventricles. Mice with a tracer depot present in brain parenchymal tissue were assumed to be misinjected and were excluded from further analysis.

In initial experiments, we euthanized groups of mice after 10, 30, and 60 min ($n = 5$ each) to determine the pattern of spread of P40D680. As early as 10 min after the infusion, we observed that

a portion of the intraventricularly infused tracer appeared to localize along the paravascular spaces of arteries within the SAS at the circle of Willis at the base of the brain with spreading within this space along the middle cerebral artery and its branches over the convexities of the cortical hemispheres (Fig. 1a–c). With increasing time, the filling of the paravascular spaces became more extensive with almost complete coverage of the blood vessels, both arteries and veins, apparent at 1 h after infusion (Fig. 1d–g). There was also entry of tracer into the Virchow–Robin spaces around penetrating arteries (Fig. 1h), similar to the "glymphatic flow" described by Iliff et al.[5]

**Lymphatic routes of CSF from the cranial cavity.** After euthanizing mice at 10, 30, and 60 min after infusion, we assessed the dynamics of the P40D680 signal within the deep cervical and mandibular lymph nodes of the neck. Deep cervical nodes already had signal at 10 min after the completion of the infusion, as shown in representative images of the left cervical nodes at this time point in Fig. 2a. The signals steadily increased with time, with significantly more tracer found in the nodes at 60 min when compared to 10 min (25,916 ± 10,741 vs. 7723 ± 3333 counts, $p < 0.05$; one-way ANOVA, Fig. 2b). Although the mandibular lymph nodes had slightly delayed dynamics, all mice had tracer within the nodes at 30 min after infusion (Fig. 2c). A significantly increased amount of tracer was found at 60 min compared to 10 min after infusion (28,544 ± 11,942 vs. 2382 ± 1214 counts, $p < 0.01$; one-way ANOVA, Fig. 2d). These findings suggest that lymphatic outflow from the CSF is rapid and sustained during the time points examined.

We next aimed to characterize the lymphatic outflow routes from the murine skull. As perineural outflow pathways have been described in other species, we looked for P40D680 tracer near cranial nerve exit routes from the skull at 60 min after intraventricular infusion in Prox1-GFP mice[14,16]. Examination of the base of the skull demonstrated accumulation of the tracer around the optic and trigeminal nerves (CN II and V) as they exit the skull (Fig. 3a, c). Tracer was also apparent at the cribriform

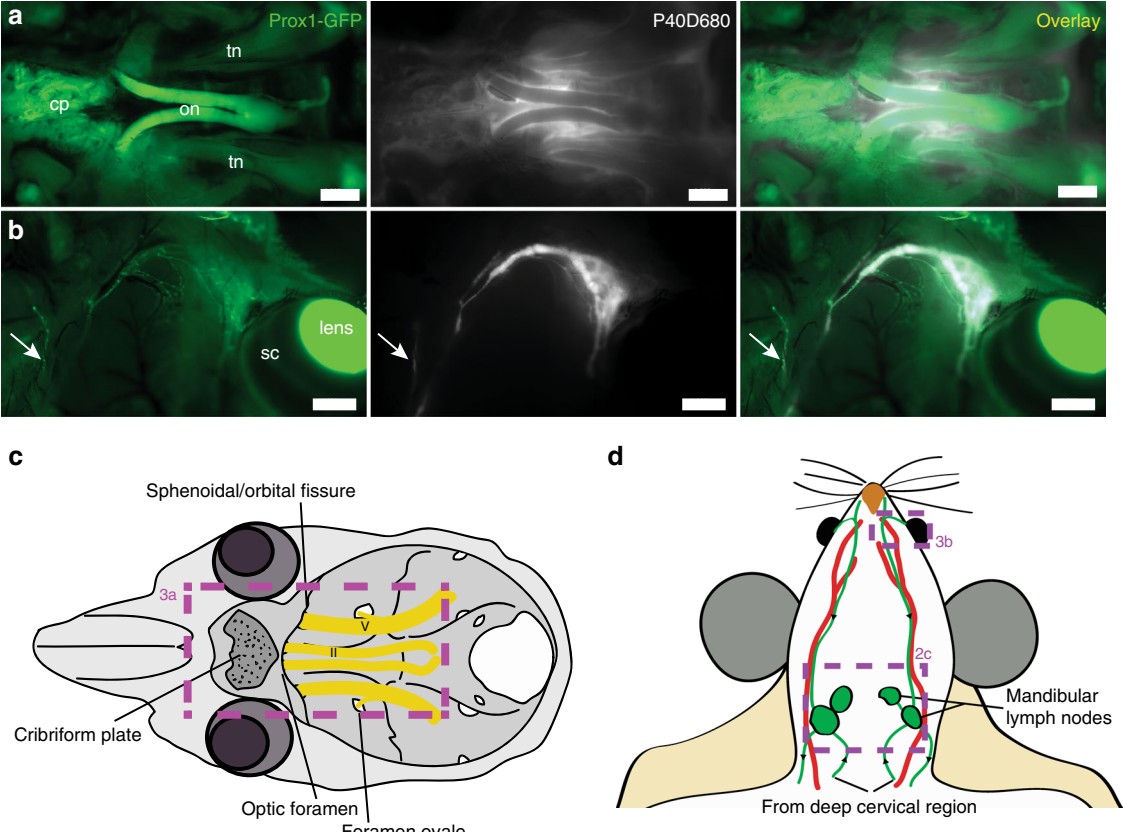

**Fig. 3** Perineural CSF outflow pathways to mandibular lymph nodes. **a** Representative image showing the perineural outflow at the base of the skull after removal of the brain. tn: trigeminal nerve, on: optic nerves, cp: cribriform plate. Scale bars: 1 mm. **b** Representative image showing the lymphatic outflow from the orbit with skin around the eye removed. Prox1-GFP expression is obvious in the lens, Schlemm's canal (sc), and tracer-filled lymphatic vessels draining the orbit. Arrow indicates a tracer-filled lymphatic from the nasal region. Scale bars: 1 mm. Images are acquired at $t = 60$ min after infusion and are representative of $n = 10$ mice. **c** Scheme of the base of the skull with the optic (II) and trigeminal (V) nerves with associated foramina and the cribriform plate indicated. Box indicates region of image in **a**. **d** Scheme outlining the CSF outflow routes to the mandibular lymph nodes. Green: lymphatic vessels with black arrows indicating direction of flow, red: facial arteries. Veins are omitted for clarity. Boxes indicates regions of images in **b** and in Fig. 2c

plate that separates the cranial and nasal cavities, indicating that outflow had also occurred along the olfactory nerves (CN I). Removal of the skin around the eye allowed visualization of a dense plexus of tracer-filled lymphatic vessels exiting the orbit that coalesced into one collecting vessel coursing along the anterior facial vein towards the mandibular lymph nodes (Fig. 3b, d). There were often vessels that joined into or ran along this collecting vessel that originated from the nasal region. By extraction of the lower jaw and removal of the tongue, esophagus, and trachea, we were able to observe bright signal in the nasal cavity through the palate and tracer emanating from a plexus of lymphatic vessels on the pharynx, which tracked towards the deep cervical lymph nodes (Fig. 4a, d). We exposed the deep cervical region and observed tracer-filled lymphatic vessels stemming from the jugular foramen on the medial side of the tympanic bulla close to where the glossopharyngeal (CN IX), vagus (CN X), and accessory (CN XI) nerves emerge (Fig. 4b, d). These vessels then proceeded toward the deep cervical lymph nodes. An additional location that tracer appeared to flow out of the skull was found on the lateral side of the tympanic bulla at the stylomastoid foramen where the facial nerve (CN VII) exits (Fig. 4c, d). Drainage from this exit point proceeded towards both mandibular and deep cervical lymph nodes. There was extreme variability between mice in the collecting vessels in the deep cervical region (Supplementary Fig. 2), with several afferent collecting vessels tracking towards up to three deep cervical lymph nodes on each side. In some cases, it appeared as though

collecting vessels were skipping the deep cervical node and that there were also connections from the deep cervical region to the mandibular nodes. A cartoon demonstrating the typical lymphatic outflow routes for tracer in the deep cervical region is shown in Fig. 4d.

As dural lymphatic vessels have been proposed to drain CSF within the cranial cavity, we also examined the lymphatic vessels along the superior sagittal and transverse sinuses within the meninges adhered to the dorsal aspect of the skull[30,31]. These vessels were easily identifiable in situ by their Prox1-GFP signals. At all the examined time points, we were unable to observe any filling of the P40D680 tracer into these vessels despite the obvious presence of tracer in the SAS of these regions (Supplementary Fig. 3a–d). Additional lymphatic vessels were found in proximity to the optic nerve at the base of the skull. These vessels also did not show obvious tracer uptake, despite clear perineural localization of tracer in the immediate vicinity (Supplementary Fig. 3e). Indeed, the lymphatic vessels generally had very small diameters and often formed a discontinuous network (Supplementary Fig. 3f–h).

**Lymphatic outflow of a macromolecular tracer from CSF.** Quantification of lymphatic outflow of CSF has been attempted in many species using approaches such as recovery of radiolabeled tracers through cannulation of collecting lymphatic vessels or detection of signal in draining lymph nodes with imaging

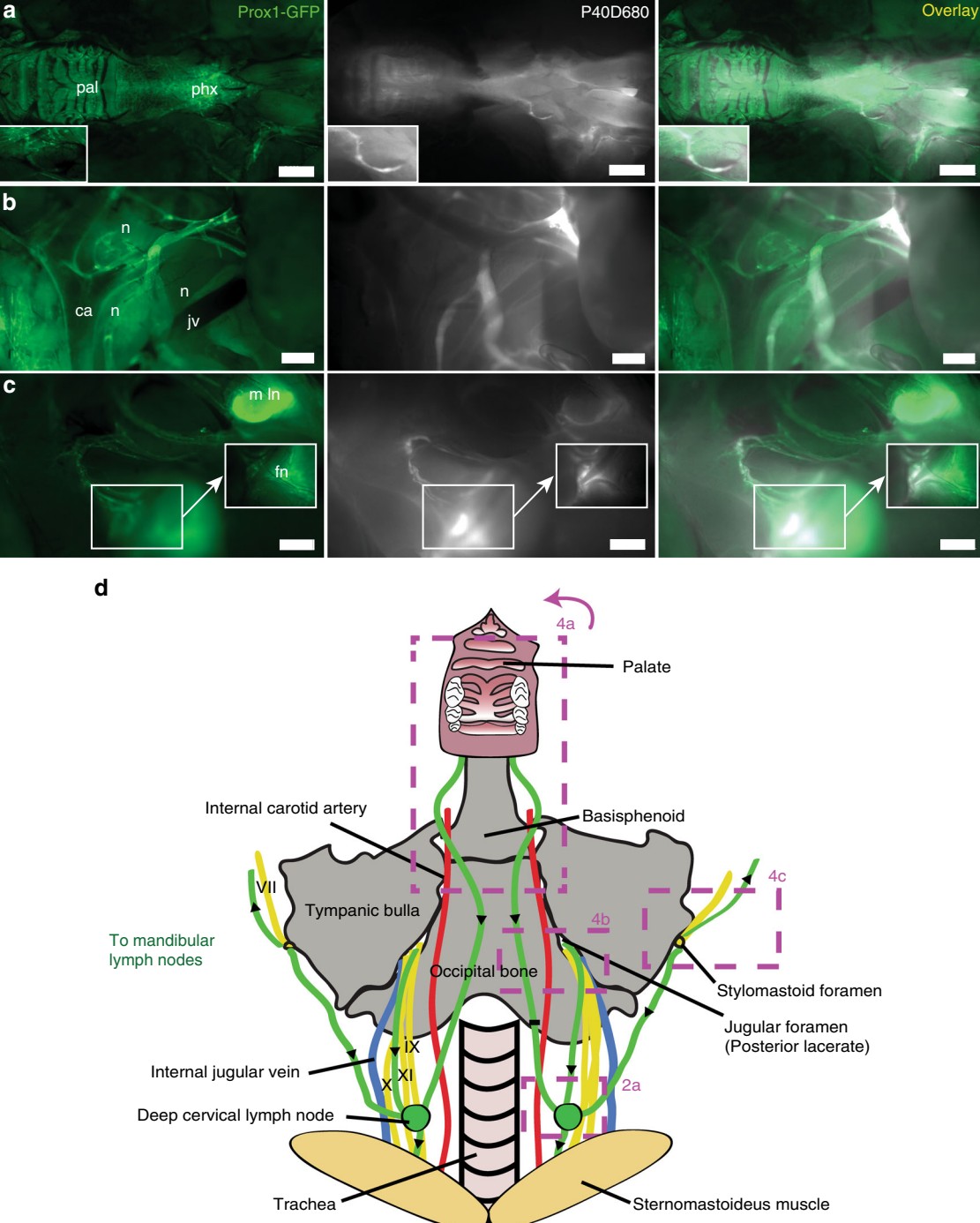

**Fig. 4** CSF outflow pathways to deep cervical lymph nodes. **a** Representative image showing lymphatic outflow from the nasal cavity. Lower jaw, tongue, trachea, and esophagus have been removed. pal: palate, phx: pharynx. Inset shows a tracer-filled collecting lymphatic vessel. Scale bar: 2 mm. **b** Representative image showing outflow from the jugular foramen on the medial side of the tympanic bulla. n: cranial nerves IX, X, and XI, jv: jugular vein, ca: carotid artery. Scale bar: 500 μm. **c** Representative image showing the perineural outflow along the facial nerve (fn) from the stylomastoid foramen on the lateral side of the tympanic bulla towards a mandibular lymph node (m ln). Scale bar: 1 mm. Images are acquired at $t = 60$ min after infusion and are representative of $n = 10$ mice. **d** Scheme demonstrating the outflow routes in the deep cervical region. Green: lymphatic vessels with black arrows indicating direction of flow, yellow: cranial nerves, red: arteries, blue: veins. Boxes indicates regions of images in **a–c** and in Fig. 2a

approaches[18,19,36,37]. Our above findings have revealed that the multiple routes of lymphatic flow and the anatomical variation between animals indicate that such previous approaches would be limited. Therefore, we aimed to apply a technique recently developed in our lab, which allows quantification of lymphatic transport from an organ by measurement in a peripheral blood vessel of an interstitially injected tracer[35]. This approach assumes

that the tracer is lymphatic-specific for the tissue of interest and that the tracer is not retained in the tissue or within draining lymph nodes by phagocytosis. The P40D680 tracer that we have used in the anatomical studies was shown to be highly sensitive for this assay with a direct linear relationship between the amount of intravenously infused tracer and saphenous vein signal, and a detection threshold of ~0.2% of the injected dose in blood

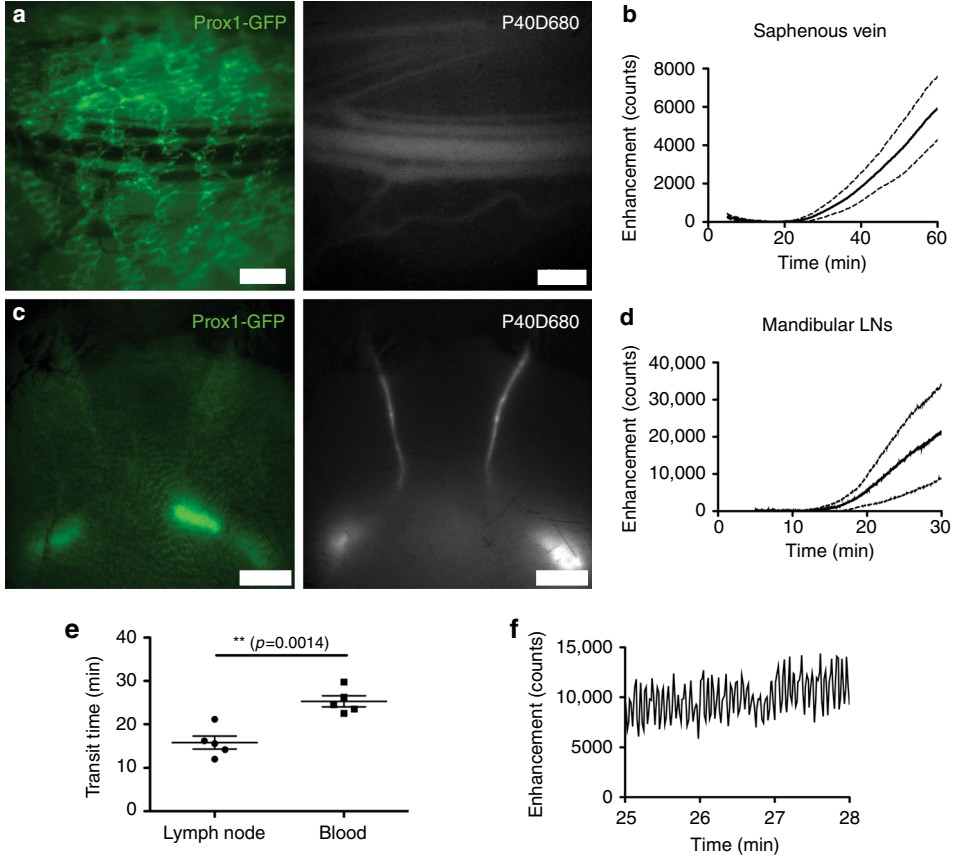

**Fig. 5** Dynamics of CSF outflow to systemic blood and mandibular lymph nodes. **a** Representative GFP and P40D680 images of the saphenous vein region in a Prox1-GFP mouse 60 min after lateral ventricle infusion of P40D680. Prox1-GFP+ dermal lymphatic vessels can be observed overlying the saphenous bundle of blood vessels. Scale bars: 500 μm. **b** Saphenous vein signal enhancement plot of $n = 5$ mice showing delayed tracer transport to systemic blood after lateral ventricle infusion of P40D680. The slight loss of signal at 5–10 min is due to photobleaching of endogenous autofluorescence (solid line: mean value, dashed line: SD). **c** Representative GFP and P40D680 images of the superficial aspect of cervical region (region shown in Fig. 3d) in a Prox1-GFP mouse 30 min after lateral ventricle infusion of P40D680. Prox1-GFP+ mandibular lymph nodes can be observed through the skin and a subset of these nodes are filled with P40D680. Scale bars: 2 mm. **d** Signal enhancement plot of mandibular lymph nodes (LNs) in $n = 5$ mice (solid line: mean value, dashed line: SD). **e** Quantification of transit time of P40D680 tracer to mandibular lymph nodes and systemic blood after intraventricular infusions ($n = 5$ each). **p < 0.01 (two-tailed Student's t-test). Data are mean ± SD. **f** Representative plot of signal enhancement in afferent collecting lymphatic vessels of the mandibular lymph nodes at a time point of 25–28 min after infusion showing a contractile pattern

(Supplementary Fig. 4). We therefore tested whether we could quantify outflow from the CSF after intraventricular infusion using this technique.

We infused 2.5 μL of P40D680 into the right lateral ventricle and after 5 min initiated noninvasive imaging of the saphenous vein with NIR stereomicroscopy (Fig. 5a). The venous signal pattern demonstrated an approximate 25 min delay before signal was seen and a steady increase thereafter (Fig. 5b; Supplementary Movie 1). This pattern is indicative of lymph outflow as previously observed in the subcutaneous tissue and the peritoneal cavity[35]. Unlike these tissues, however, outflow was robust from the CSF even under anesthesia, with an estimated 21.6% ± 6.0% ($n = 5$ mice) of the infused dose within the bloodstream at 60 min based on the calibration curve shown in Supplementary Fig. 4a.

If the delay in the initial signal increase in the saphenous vein was due to lymphatic outflow then functional lymphatic transport should be apparent at earlier time points. As the mandibular lymph nodes are located directly under the skin, we speculated that we would be able to noninvasively visualize these nodes with NIR imaging similar to a previous report[38]. As before, we infused P40D680 into the lateral ventricle and placed the Prox1-GFP mice under the stereomicroscope with a region of interest on the shaved neck skin (Fig. 5c). We could easily noninvasively

visualize tracer transport within collecting lymphatic vessels and into the lymph nodes as early as 11 min after the infusion (Fig. 5d; Supplementary Movie 2). In $n = 5$ mice that were imaged with each protocol, we detected an average initial signal increase at 15.9 ± 3.4 min (transit time to nodes) in the draining lymph nodes, which was significantly earlier than the increase at 25.3 ± 2.8 min representing the transit time to blood (Fig. 5e). Active contractility of the afferent collecting lymphatic vessels of the mandibular lymph nodes was also observed with contraction frequencies of 13.88 ± 2.36 per min (Fig. 5f). Although they could not be visualized in a similar manner noninvasively, the deep cervical lymphatic trunks were also transporting fluid at early time points as signal increases were often detected in videos in the more caudal mandibular nodes before superficial afferent vessels were observed, demonstrating a functional connection from the deeper network. In addition, as shown in Fig. 2b above, deep cervical node signal was apparent in situ in all mice when euthanized 10 min after the infusion.

The delay in the time for the signal to be apparent in the blood suggested that there did not appear to be rapid venous uptake of the tracer, implying that direct routes into the blood may not be active under these conditions. To provide further evidence for this, we tested whether we could detect a direct blood outflow of

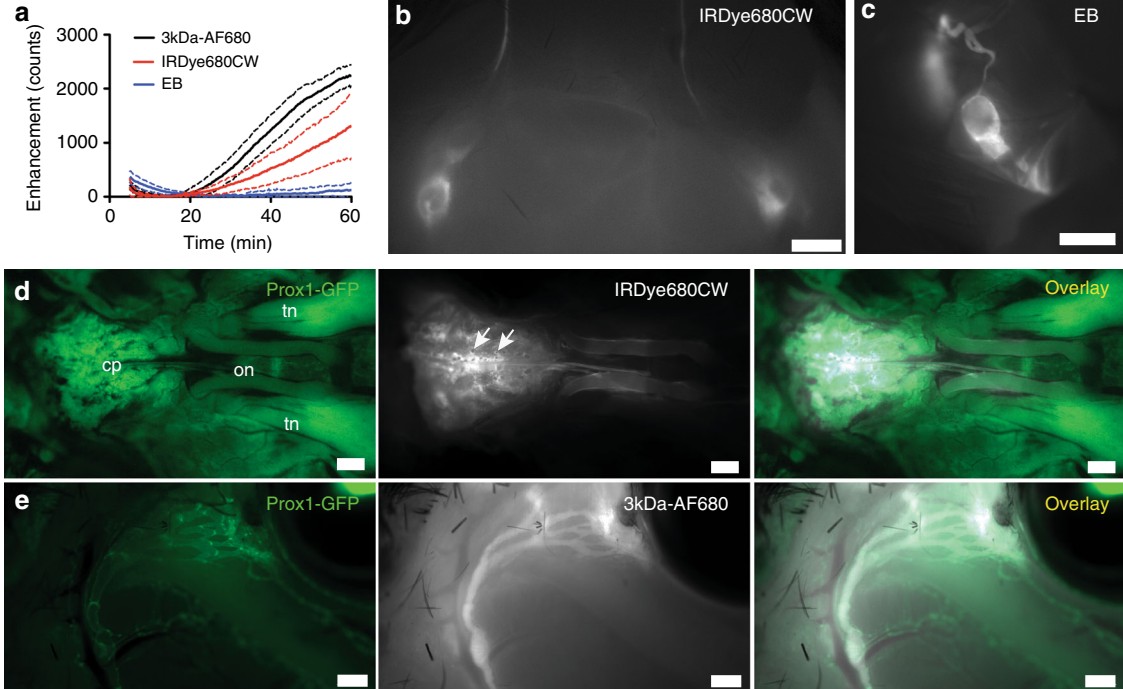

**Fig. 6** CSF outflow after intraventricular infusions of small molecular tracers. **a** Saphenous vein signal enhancement plots of Prox1-GFP mice after lateral ventricle infusions of 0.6% EB, 200 µM IRDye680CW, or 1.6 mg/mL 3kDa-AF680 ($n = 4$ each; solid line: mean value, dashed line: SD). The loss of signal from 5 to 15 min is due to photobleaching of endogenous autofluorescence. **b** Representative image of mandibular lymph nodes and afferent collecting vessels 60 min after lateral ventricle infusion of 200 µM IRDye680CW. Similar images were acquired for EB and 3kDa-AF680. Scale bar: 1 mm. **c** Representative image of the left deep cervical lymph node and collecting vessels 60 min after lateral ventricle infusion of 0.6% Evans blue. Similar images were acquired for IRDye680CW and 3kDa-AF680. Scale bar: 1 mm. **d** Representative image showing the perineural outflow of IRDye680CW at the base of the skull after removal of the brain. tn trigeminal nerve, on optic nerves, cp cribriform plate. White arrows indicate foramina in the cribriform plate. Scale bars: 500 µm. Similar images were acquired for EB and 3kDa-AF680. **e** Representative image showing the lymphatic outflow of 3kDa-AF680 from the orbit with skin around the eye removed. Scale bars: 500 µm. Similar images were acquired for EB and IRDye680CW

the tracer by imaging the posterior facial vein (Supplementary Fig. 5a). In rodents, the major venous outflow routes for blood from the brain and the dural sinuses exit the skull through the postglenoid (or "spurious" jugular) foramina and drain into the posterior facial veins to reach the external jugular veins, as opposed to the internal jugular veins which act in this capacity in larger mammalian species including humans[39–41]. Infusions into the lateral ventricle were performed in $n = 5$ mice and an incision of the skin on the lateral aspect of the head was made to expose the posterior facial vein. Imaging was initiated at 10 min after the completion of the infusion and continued for 20 min. At no point was tracer apparent within the posterior facial vein (Supplementary Fig. 5b, c; Supplementary Movie 3) until around 25 min after infusion at which point an increase in signal was detected. The time point of this signal increase and the overall signal enhancement at 30 min were directly comparable to the values detected at the saphenous vein, indicating that a systemic blood signal increase had occurred. We validated at this point that there was signal within the saphenous vein as well as within the lymphatic vessels leading to the mandibular and cervical lymph nodes (Supplementary Fig. 5d–f). On the basis of our dynamic imaging data, we conclude that lymph transport was the main route for tracer movement from the CSF into the systemic bloodstream rather than direct blood outflow through the posterior facial vein.

### CSF outflow of small molecules through the lymphatic system.

We next determined whether small molecular tracers would exhibit similar outflow patterns compared to P40D680. Three fluorescent small molecular tracers with excitation and emission

wavelength properties similar to P40D680 were utilized: Evans blue (EB), IRDye680CW, and a 3 kDa dextran conjugated to AlexaFluor680 (3kDa-AF680). Initial experiments were performed to establish comparable doses for intraventricular infusion as well as to test the sensitivity of detection in the saphenous region. It was observed during these studies that although both IRDye680CW and 3kDa-AF680 were not retained within the blood compartment after intravenous injection, they leaked rapidly in sufficient quantities within the skin above the saphenous vein to be detected with high sensitivity (Supplementary Fig. 6). In groups of $n = 4$ mice, we infused 2.5 µL of either 0.6% EB, 200 µM IRDye680CW, or 1.6 mg/ml 3kDa-AF680 into the right lateral ventricle and initiated imaging of the saphenous vein. Surprisingly, we could not detect immediate blood uptake of any tracer, with the patterns exhibiting delays before signal could be detected peripherally (Fig. 6a). After euthanization, we confirmed that lymphatic outflow of the tracers was occurring with similar routing to lymph nodes as P40D680 (Fig. 6b, c). With the small molecules, obvious transport through foramina of the cribriform plate occurred with strong signals in this region (Fig. 6d). There was also signal apparent around the optic nerves (Fig. 6d) with outflow from the orbit similar to P40D680 (Fig. 6e). Noticeably, less EB reached the bloodstream compared to the other two small molecular tracers. Upon examination of brain sections, EB appeared to directly enter the parenchyma of the brain through the ependymal lining of the ventricles (Supplementary Fig. 7a) in more significant quantities than IRDye680CW and 3kDa-AF680. Fluorescence imaging indicated more significant accumulation of EB and IRDye680CW around arteries compared to veins (Supplementary Fig. 7b), rather than the paravascular spreading

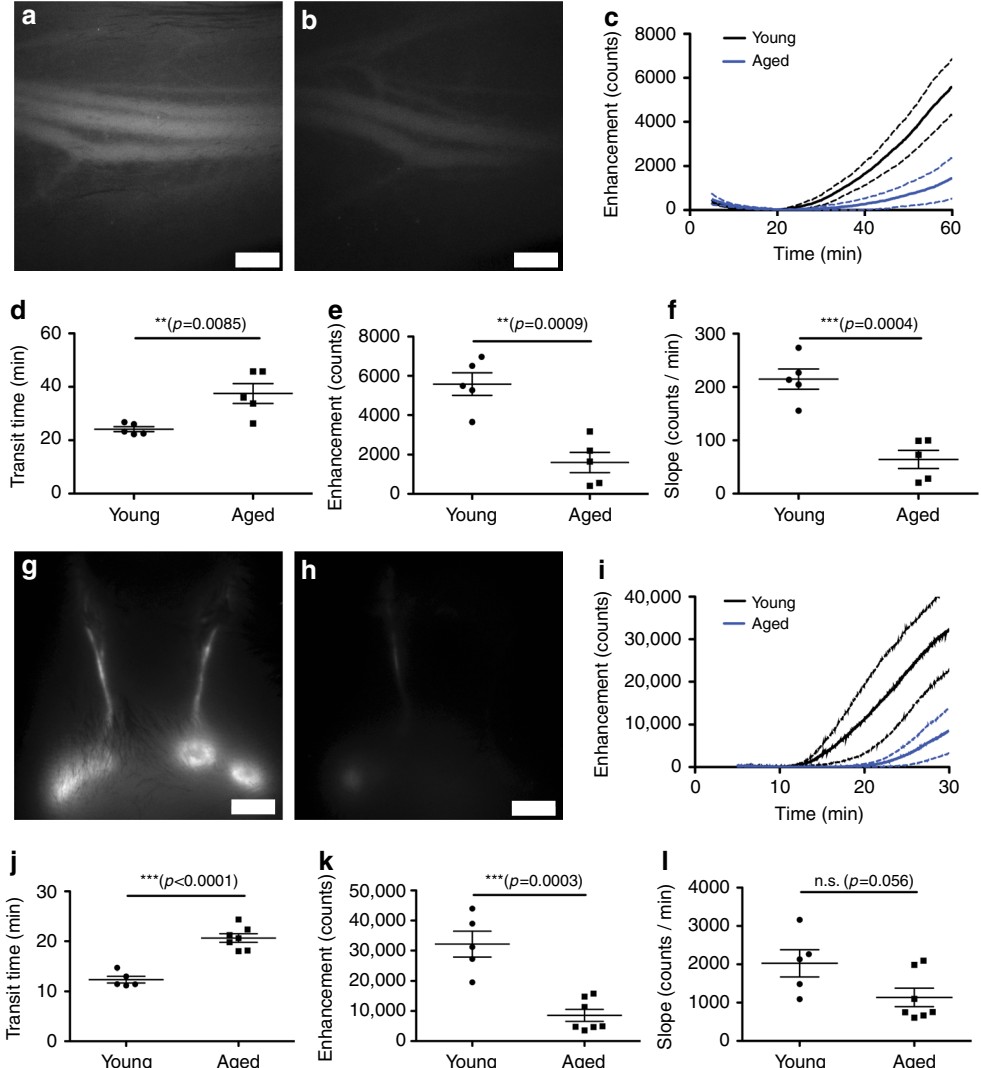

**Fig. 7** CSF outflow to systemic blood and lymph nodes in young and aged mice. **a** Representative image of the saphenous vein in a young (2-month-old) mouse 60 min after lateral ventricle infusion of P40D680. **b** Representative image of the saphenous vein in an aged (18-month-old) mouse at the same time point. Scale bars: 500 μm. **c** Saphenous vein signal enhancement plots of young and aged mice ($n = 5$ each). Solid line indicates mean, dashed lines indicate SD. **d** Quantification of transit time. **e** Quantification of signal enhancement at 60 min. **f** Quantification of slopes of the signal enhancement from 45 to 60 min. **p < 0.01, ***p < 0.001 (two-tailed Student's t-test). Data are mean ± SD. **g** Representative image of the mandibular lymph node region in a young mouse 30 min after lateral ventricle infusion of P40D680. **h** Representative image of the mandibular lymph node region in an aged mouse 30 min at the same time point. Scale bars: 2 mm. **i** Mandibular lymph node enhancement plots of young ($n = 5$) and aged mice ($n = 7$). Solid line indicates mean, dashed lines indicate SD. **j** Quantification of the transit time to the mandibular lymph nodes. **k** Quantification of average signal enhancement at the lymph nodes at 30 min. **l** Quantification of the average slopes of the signal enhancement in the lymph nodes during the last 5 min of imaging. **p < 0.01, ***p < 0.001 (two-tailed Student's t-test). Data are the mean ± SD

pattern that was observed around both blood vessel types with P40D680. Both EB and IRDye680CW appeared to bind strongly to the arteries surrounding the SAS (Supplementary Fig. 7c). On the other hand, 3kDa-AF680 showed less specific binding to arteries with distribution of the tracer within regions between arteries and veins of the pia mater. In summary, the results indicate that despite evidence of varying degrees of retention of these small molecular tracers within the CNS, the CSF outflow routes through lymphatics and the dynamics of transport to systemic blood were similar to those of a macromolecular tracer.

**Reduced lymphatic outflow from CSF in aged mice.** The incidence of Alzheimer's disease and other dementias and many other neurodegenerative diseases increases with age. Recently, a

hypothesis for the development of these disorders was proposed that toxic proteins such as amyloid beta and tau may accumulate in the brain due to reduced clearance[42]. Earlier studies have shown a reduced turnover of CSF and removal of labeled proteins, including amyloid beta, after ventricular-cisternal perfusion in aged rats[43]. Therefore, we tested whether lymphatic outflow from CSF was reduced in aged mice and would be similar to prior observations of reduced peripheral lymphatic transport in aged mice[34,35,44]. We first tested whether we could detect differences in CSF transport to blood after ventricular infusion of P40D680 in 18-month-old (aged) and 2-month-old (young) C57BL/6J-Tyrc-J albino mice. After 60 min, we observed less P40D680 signal in the saphenous vein of aged mice compared with the young controls (Fig. 7a, b). Dynamic imaging (Fig. 7c; Supplementary Movies 4 and 5) showed a significant increase in the transit time to blood in

aged mice ($37.5 \pm 8.4$ vs. $24.2 \pm 2.1$ min, $p = 0.0085$; $t$-tailed Student's $t$-test, Fig. 7d) compared with young mice. The signal enhancement at 60 min in aged mice was significantly reduced compared with young mice ($1600 \pm 1160$ vs. $5580 \pm 1285$ counts, respectively; $p = 0.0009$; two-tailed Student's $t$-test, Fig. 7e), as was the slope of enhancement from 45 to 60 min ($64.1 \pm 38.0$ vs. $214.9 \pm 42.5$ counts/min, $p = 0.0004$; two-tailed Student's $t$-test, Fig. 7f). These results indicate that the dynamics of CSF outflow to blood were slower in aged mice.

We next tested whether less lymphatic outflow from CSF to the submandibular lymph nodes also occurred in aged mice. As before, we infused P40D680 into the right ventricle and imaged the submandibular lymph node region through the skin. There was less signal seen in the lymph nodes of aged mice at 30 min compared to young controls (Fig. 7g, h). Quantification from dynamic imaging (Fig. 7i) revealed a significantly increased transit time to the lymph nodes in aged mice compared to young mice ($20.7 \pm 2.3$ vs. $12.4 \pm 1.5$ min, $p < 0.0001$; two-tailed Student's $t$-test, Fig. 7j). Signal detected in the mandibular nodes at 30 min was markedly reduced (aged: $8567 \pm 5273$ vs. young: $32173 \pm 9617$ counts; $p = 0.0003$; two-tailed Student's $t$-test, Fig. 7k). The slope of signal increase during the last 5 min of imaging (aged: $1136 \pm 637$ vs. young: $2027 \pm 795$ counts/min; $p = 0.056$; two-tailed Student's $t$-test, Fig. 7l) was not significantly different. Further analysis after euthanizing the mice showed that less signal was also apparent in the deep cervical lymph nodes at 30 min in aged mice (aged: $7780 \pm 2781$ vs. young: $15123 \pm 4756$ counts; $p = 0.006$; two-tailed Student's $t$-test, Supplementary Fig. 8). However, no significant difference could be seen at 60 min indicating a delayed outflow pattern in aged mice.

As the delay of outflow that was observed after intraventricular infusion could possibly be due to altered transport from the lateral ventricle to the SAS in aged animals, we also tested for a reduction in lymphatic outflow in aged mice after cisterna magna infusion. Cisterna magna infusions of 5 µL of 100 µM P40D680 were performed in 18-month-old (aged) and 2-month-old (young) C57BL/6J-Tyrc-J albino mice. The infusion cannula was kept in place for 10 min to limit the backflow of tracer from the CSF and imaging was initiated at 15 min after the completion of the infusion. Once again, aged mice demonstrated a reduction in CSF outflow with lower signals apparent in the systemic blood at 60 min (Supplementary Fig. 9). Quantification revealed an increased transport time, decreased signal in blood at 60 min, and a lower slope of enhancement from 45 to 60 min in aged mice compared to younger controls. In sum, the imaging data after both intraventricular and cisternal infusions indicated significantly slower CSF outflow into lymphatic vessels of aged mice.

## Discussion

We have elucidated a complex flow pattern of CSF after ventricular infusion of tracers in mice, demonstrating various perineural outflow routes from the mouse cranial cavity, as well as flow from the ventricles to the SAS and into the nervous tissue of the brain via paravascular spaces. Dynamic imaging suggested that lymphatic outflow was the major outflow route for CSF rather than venous routes, with even small molecular tracers not demonstrating immediate blood uptake. Finally, we have demonstrated a significant decline in CSF outflow in aged mice, a finding that may have significance for many aging-associated neurological disorders.

Our study is, to our knowledge, the first to characterize in detail the lymphatic outflow pathways of CSF from the skull in mice (Supplementary Fig. 10). Our use of pegylated and small molecular NIR tracers and lymphatic-specific reporter mice combined with high-resolution stereomicroscopy have allowed an in-depth analysis of perineural outflow pathways and routing to CSF-draining lymph nodes. We found an extensive network of lymphatic vessels outside the CNS that transported tracer from the CSF to both mandibular and deep cervical lymph node groups within 30 min after intraventricular infusion. Our findings of perineural outflow largely agree with a wide-ranging body of research in many species dating back to Schwalbe in 1869 that detail the major outflow pathways from the skull to reach the lymphatic system[15,17]. Most previous researchers have come to the conclusion that perineural drainage along the olfactory nerve (CN I) through the cribriform plate to reach the nasal mucosa is the most important CSF lymphatic outflow route in several species[17,20,21,45]. Although we were able to confirm that this pathway was active, we also highlighted several other perineural routes that appear to complement the nasal route in mice. In agreement with previous literature, the perineural pathways along the optic (CN II) nerve through the orbit appeared to be important[22–24]. However, we have shown that the trigeminal nerve (CN V) also exhibited a perineural pattern and, therefore, may represent another pathway from the cranial SAS to the orbit. We also detected tracer emanating from the jugular foramina along the glossopharyngeal (CN IX), vagus (CN X), and accessory (CN XI) nerves, as well as outflow from the stylomastoid foramina along the facial nerve (CN VII), two locations that are in close proximity to the deep cervical lymph nodes. Interestingly, jugular outflow was detected in the early study in dogs by Schwalbe, however, these specific pathways through the base of the skull have otherwise been identified in only a few previous studies[15,29,46,47]. In sum, we found evidence of tracer exiting along several cranial nerves in mice after intraventricular infusion.

The recent rediscovery of dural lymphatic vessels has attracted a great deal of attention on a role for a functional lymphatic system within the central nervous system[30,31]. Although these vessels have been previously observed in many species, these earlier findings have been largely ignored[21,48–51]. In the current study, we were not able to observe any apparent uptake of the P40D680 tracer into the dural lymphatic vessels within the skull near the superior sagittal and transverse sinuses, or near the optic nerve at several different time points after infusion. We found that these lymphatic vessels were frequently discontinuous, had small diameters and, as previously reported[30,31], lack intraluminal valves (except at the base of the skull), indicating that they are different from conventional lymphatic networks. One possible objection to a functional role for these vessels draining CSF is the presence of an arachnoid barrier layer between the CSF and the dura mater[4,52,53]. Similar to the barriers at the endothelium of the brain capillaries and at the epithelial lining of the choroid plexus[3], there exists a barrier layer that consists of arachnoid cells containing tight junctions that serves to isolate the CSF within the SAS from the interstitial fluid of the dura mater[4,52,53]. As the blood vessels of the dura mater tissue are fenestrated[4], it is possible that the dural lymphatic vessels exist to drain this tissue. A previous study found the presence of horseradish peroxidase in dural lymphatic vessels of the cat only under high injection pressures indicating that physical disruption of the arachnoid barrier must occur in order for CSF to reach the dura mater tissue[50]. Others have discussed that dural lymphatics must have a minor or accessory role to the main perineural outflow pathways[21,54]. Interestingly, anatomical studies have demonstrated lymphatic vessels in the dura around the optic and facial nerves and at the cribriform plate and jugular foramen, indicating that lymphatic uptake of CSF may occur as the cranial nerves exit the skull[31,49,55,56]. Clearly, more research is needed to determine the importance of the dural lymphatic route for CSF outflow in comparison to the more established perineural pathways to reach extracranial lymphatic vessels.

Our development of NIR dynamic imaging approaches has allowed real-time quantitative assessments of both lymphatic outflow and transport to blood of CSF-infused tracers that were not previously possible in animals without cannulation. The dynamic imaging results after CSF infusions of macromolecular and small molecular fluorescent tracers strongly suggest that lymphatic transport is the predominant CSF outflow pathway in mice. Supporting evidence for this conclusion is (1) the similarity of the CSF outflow pattern to patterns of lymphatic transport to blood from other tissues such as skin and peritoneal cavity[35], with a delay in the appearance of the signal in blood followed by a steady linear increase; (2) the presence of significant levels of tracer within the lymphatic vessels and lymph nodes draining the skull before any appearance of signal in the blood; (3) the lack of any evident signal during direct imaging of the posterior facial vein which receives the blood from the transverse sinus in rodents[39,40]; and (4) the confirmation of an active pumping mechanism of collecting lymphatic vessels draining CSF that serves to propel the lymphatic fluid towards the venous system even under anesthesia. These results, indicating a lymphatic-predominant drainage of CSF, are largely in agreement with work from the group of Miles Johnston who reached a similar conclusion[54,57]. Starting in the late 1990s, a series of elegant quantitative studies in sheep and rats from this group demonstrated that radiolabeled tracers could be recovered in sufficient quantities from cannulated lymphatic vessels to account for over half of the tracers estimated within the bloodstream[17,19]. The robust outflow through lymphatic vessels under anesthesia observed in the current study is likely due to a favorable pressure gradient driving flow from the SAS within the rigid cranial cavity to the lymphatic vessels outside the skull. Measurements in other species have indicated that pressure within the SAS may be 3 or more times that of the cervical lymphatic system[58]. Although we have shown previously that lymphatic transport is reduced under anesthesia due to lack of uptake into initial vessels[35], the pressure gradient between the skull and surrounding tissue may overcome the need for muscular movement that normally drives lymph formation. Nonetheless, the question whether CSF lymphatic outflow is increased or decreased during awake conditions remains to be answered.

These findings call into question the role of arachnoid villi or other possible direct venous connections as outflow routes of solutes from the CSF. It is perhaps not surprising, due to their location within the skull, that no direct in vivo proof of arachnoid villi uptake has ever been demonstrated. Supporting evidence for the importance of these structures draining CSF have mostly relied on examination of post-mortem tissue[6,59] or on ex vivo studies, which have shown in isolated arachnoid villi that particles as big as erythrocytes could pass through[60]. However, since a continuous lining of endothelial cells with tight junctions was found to exist on the villus[8,11], it is still unclear how macromolecules are transported[7,12,61]. Previous studies have attempted to use the dynamics of accumulation of intraventricular-injected macromolecular tracers into the blood or urine as evidence of arachnoid villi transport[62–64]. In these studies there were often delays until tracer was apparent in the blood or urine compared to when the tracers were injected intravenously, similar to our findings of a delayed transport to blood. We were also unable to provide evidence of a direct route from the CSF to blood by dynamic imaging of the posterior facial vein that collects blood from the transverse sinus in mice within the first 30 min after infusion despite the presence of high levels of tracer within lymphatic vessels. These results are similar to an early cannulation study in dogs[65]. Studies from the group of McComb under various CSF pressure conditions were unable to detect increased tracer levels in the superior sagittal sinus compared to peripheral

blood in rabbits, cats, and primates[64,66,67]. In rats, direct blood uptake was detected after intraventricular infusions of inulin or polystyrene beads only after reaching CSF pressures of seven to eight times normal levels[68]. Studies in sheep by the Johnston group determined that pressures of around threefold higher than normal levels were necessary to detect direct blood uptake[13]. Therefore, it appears that plenty of doubt exists regarding the role of the arachnoid projections in the drainage of fluid and macromolecules from the CSF in many species beyond mice, and that it may be time for a reexamination of this widely accepted concept. Although the present experiments cannot explicitly rule out a direct connection from CSF to blood, particularly one that is active at later time points or under high intracranial pressure, they are strongly indicative that the lymphatic outflow pathways are much more important than originally envisioned and that the lymphatic system may represent the major site between the CSF and blood compartments.

To our knowledge, our study is the first to demonstrate a reduction of lymphatic outflow of CSF with aging. One previous study has shown less outflow of radiolabeled tracers to the nasal turbinates in aged rats but did not quantify outflow to the lymphatic system or the systemic blood[69]. Since under homeostatic conditions, CSF inflow must equal CSF outflow, this decreased outflow in aged mice may be at least in part due to reduced CSF production by the choroid plexus or by sources from within the interstitial tissue such as the blood–brain barrier[16,70]. The choroid plexus has been shown to undergo age-related morphological alterations in its epithelial lining as well as a thickening of the basement membrane[43]. An additional explanation for the reduced CSF outflow observed during the current study could be previous data from humans and rats showing an increased CSF volume with aging, which would indicate that the infused tracer may have been more diluted in the CSF of aged mice[43,71]. It will be interesting to test whether a more accelerated decrease in CSF lymphatic outflow exists in mouse models of Alzheimer's disease and whether a functional decline is associated with the development of amyloid beta plaques. If so, the lymphatic system may represent a possible new therapeutic target with the aim to enhance the clearance of toxic proteins from the CSF and the brain.

In conclusion, we have identified many perineural sites for the egress of CSF from the cranium of mice. Most importantly, through NIR dynamic imaging, we have presented evidence that these outflow pathways to reach the lymphatic system appear to represent the major exit routes for both macromolecular and small molecule tracers from the CSF, rather than the commonly accepted venous route. Although demonstration of this phenomenon in humans still awaits, this study sets the stage for investigation of a potentially important role for the lymphatic system in a multitude of CNS-related pathologies. In addition, there may be potential for clinical translation of the imaging technique since noninvasive monitoring of NIR tracers in the blood already exists in the clinic[72]. Therefore, quantification of CSF outflow after intrathecal administration of NIR tracers in patients suffering from neurological disorders may be possible.

## Methods

**Mice**. All mouse experiments were approved by Kantonales Veterinaramt Zurich (license numbers 185/13, 196/13, and 161/16). Female C57BL/6J-$Tyr^{c-J}$ albino mice (Jackson Laboratories, Bar Harbor, ME) and Prox1-GFP mice[73] on the C57BL/6J background were kept under specific pathogen-free conditions and used for experimental studies at the age of 2–3 months. For aging studies, female C57BL/6J-$Tyr^{c-J}$ mice were aged in-house to 18 months of age and compared with young animals from the same colony. After the completion of imaging experiments, the mice were euthanized with an overdose of anesthesia (1000 mg/kg ketamine; 3.5 mg/kg medetomidine).

**Infusion of tracers into lateral ventricle or cisterna magna**. Mice were anesthetized (80 mg/kg ketamine; 0.2 mg/kg medetomidine) and fixed in a stereotaxic frame (RWD, San Diego, CA). The skull was thinned with a dental drill (RWD) at a location 0.95 mm lateral and 0.22 mm caudal to the bregma. A 33 G steel needle was inserted into the right lateral ventricle 2.35 mm ventral to the skull surface. Infusion of 2.5 μL of 200 μM P40D680 tracer[32] (provided by Dr. J.-C. Leroux, ETH Zurich), 0.6% Evans blue (Sigma, St. Louis, MO), IRDye680CW NHS ester (LI-COR Biosciences, Lincoln, NE), or 3kDa-AF680 (Thermo Fischer) at the speed of 1 μL/min was then performed with a syringe pump (Stoelting, Wood Dale, IL). The needle was left in place for 2.5 min and then slowly removed while observing if any significant backflow occurred.

For cisterna magna infusion, mice were anesthetized as above and a surgical procedure to access this structure was performed[74]. After a small skin incision over the occipital bone/cervical spinal cord was performed, the three covering muscle layers were carefully dissected under a stereomicroscope using fine forceps and scissors. Subsequently, the atlanto-occipital membrane (AOM), covering the cisterna magna, was pre-punctured using a 34 G cannula. The infusion cannula was then inserted through this opening to a depth of 250 μm. A droplet of histoacryl was added to the interface of cannula and AOM to avoid reflux of CSF/tracer during the infusion. Overall, 5 μL of 100 μM P40D680 tracer was infused at the speed of 1 μL/min. After the infusion, the cannula was left in place for 10 min to avoid reflux. After withdrawal of the cannula, a droplet of histoacryl was finally added to ensure no CSF fistula.

**Dynamic NIR imaging of CSF outflow**. For noninvasive imaging of tracer signals in blood[35], fur above the saphenous vein region was shaved with a razor and depilation cream before the ventricular or cisternal infusion. After removal of the needle from the skull or the cisterna magna, the mice were quickly positioned under a Zeiss StereoLumar.V12 stereomicroscope with AxioVision software (Carl Zeiss, Feldbach, Switzerland) and a Photometrics Evolve 512 camera (Photometrics, Tuscon, AZ) in a supine position on a heating pad to maintain body temperature. The autofluorescence signal on the GFP channel was used to position the saphenous blood vessels at ×25 zoom. Dynamic imaging was initiated 5 min after the completion of the ventricle infusion and 15 min after completion of the cisterna infusion by acquisition of a sequence of images (1 image every 15 s for 55 min) with a Cy5 filter set to monitor the NIR signal of the saphenous vein. Exposure time and camera gain settings were 200 ms and 200, respectively.

For noninvasive imaging of the lymphatic flow[32] in the shaved neck region, the ventricular infusion procedure was the same as above and image acquisition was performed with the following modifications: magnification of the imaging was at ×6 zoom on the neck region and dynamic imaging was performed for 25 min with a sequence of 1 image per 1 s.

For noninvasive imaging of potential tracer outflow to the posterior facial vein, the same ventricular infusion procedure was followed, but immediately afterwards, an incision to the lateral skin of the head was made to expose the junction of the internal maxillary vein with the posterior facial vein. Dynamic imaging was initiated at 10 min after injection and continued for 20 min with 1 image acquired every 15 s at ×22 zoom.

**Assessment of transport to blood and lymph nodes**. Using AxioVision software, a circular region of interest of radius 100 μm was placed over the saphenous vein on the acquired videos. Using the Measure Profile function, a table of fluorescence intensity in counts vs. time was exported into Microsoft Excel. As there was a loss of signal at the beginning of the scans due to photobleaching of tissue auto-fluorescence, baseline intensity in counts was calculated as an average signal of the lowest ten consecutive imaging frames. This baseline intensity was then subtracted from the fluorescence intensity values to plot fluorescent signal enhancement vs. time in min. For quantification of transport to blood, three assessments were made from the data of signal enhancement vs. time. These were the NIR fluorescent signal enhancement value in counts at $t = 60$ min, the linear slope of signal enhancement from $t = 45$ to 60 min after infusion in counts/min and the transit time in min of the arrival of tracer to the blood circulation (set at a threshold of 100 counts of signal enhancement).

For assessment of transport to mandibular lymph nodes, the procedure was the same excepting that the circular region of interest was of radius 500 μm over the mandibular lymph node. The linear slope was calculated as the signal enhancement from $t = 25$ to 30 min after infusion and the threshold for signal enhancement to determine the transit time was set to 300 counts. The quantifications of one lymph node on each side were averaged to generate one value per mouse.

**Assessment of collecting lymphatic vessel contractility**. Contractility was assessed based on our published methods[33]. In short, in AxioVision software, on each video of dynamic imaging of mandibular lymph nodes, a region of interest of radius 200 μm was drawn over the afferent collecting vessel on each side of the mouse. The data in mean fluorescent intensity values over time was then exported in .xml format. Contractility measures were determined for a 5 min period between 25 and 30 min after infusion using a Matlab algorithm for the frequency in contractions per min. Values from the two vessels were then averaged to obtain one value for each measure per mouse.

**Intravenous infusions for tracer dose calibration**. A custom-designed catheter of polyethylene (PE)—10 tubing (SCI, Lake Havasu City, AZ) and a 30 G needle was placed into the tail vein of C57BL/6 J mice. An infusion pump (PHD2000, Harvard Apparatus, Cambridge, MA) connected to the catheter was used for consistent infusions of tracer. For demonstration of the linear relationship between tracer dose and venous signal, either 10 μL of the P40D680 tracer (at 1 μM) or Evans blue (at 0.003%) was infused every 3 min using a stepwise infusion program. To determine the threshold amounts for detection in blood, P40D680 was diluted to 0.05 μM and Evans blue to 0.00015%, and infused in a similar manner. The number of infusions before increasing signal was detected in the blood was determined to be the threshold for detection.

**In situ analysis of CSF flow and lymphatic outflow**. Anatomical mapping of the outflow pathways in Prox1-GFP mice was performed in three groups of $n = 5$ mice at time points of euthanization at 10, 30, and 60 min after intraventricular infusion of P40D680. Images of P40D680 tracer spread on the surfaces of the brain, peri-neural exit points, and within the lymphatic vasculature were acquired with a Zeiss AxioZoom V16 microscope and a QImaging OptiMOS sCMOS camera (QImaging, Surrey, Canada) combined with a light-emitting diode illumination system pE-4000 (CoolLED Ltd, Andover, UK). To compare signals within lymph nodes at different time points, images were acquired at identical zoom factors (×25), and exposure times (200 ms). As there were no consistent differences in signal in the lymph nodes on the injected and contralateral sides, the average value of the nodes on each side was used.

**Brain sections**. Mouse brains were dissected and fixed in 4% PFA at 4 °C overnight. A section of the brain between 1 mm cranial and 1 mm caudal to the needle insertion site was cut out by razor blade and imbedded with 2% low melt agarose (BIO-RAD). Cross sections were made from dorsal to ventral side at a thickness of 100 μm with Vibrating blade microtome (Leica VT1000 S). Images were acquired with a Zeiss AxioZoom V16 microscope.

**Statistics**. For the studies to assess paravascular spread and CSF outflow at different time points, mice were randomly allocated to different experimental groups. Owing to the nature of the aging studies, neither randomization nor blinding of investigators was possible. Group sizes were estimated based on pilot studies to determine the success rate and reproducibility of the intraventricular and cisterna magna infusions. All data are presented as mean ± SD. All groups were found to be normally distributed using the Kolmogorov–Smirnov (K–S) test. Means of two groups were compared using a two-tailed Student's $t$-test. Means of three groups were compared with one-way ANOVA with the Tukey's multiple comparison post hoc test. All analyses were performed using GraphPad Prism V5.0 (GraphPad Software, San Diego, CA) and $p < 0.05$ was accepted as statistically significant.

**Data availability**. The data that support the findings of this study are available from the authors upon reasonable request.

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

## Acknowledgements

Special thanks to Dr. Jean-Christophe Leroux and Diana Andina for the P40D680 tracer and to Dr. Martin Schwab and Dr. Markus Rudin for cantonal-approved access to animal licenses. The authors also thank Jeannette Scholl, Carlos Ochoa, Jan Salchli and Dr. Anna Polomska for excellent technical assistance and Dr. Vartan Kurtcouglu for his critical reading of the manuscript. This work was supported by Swiss National Science Foundation grants 3100A0-108207 and 31003A-130627, Advanced European Research Council grant LYVICAM, Leducq Transatlantic Network of Excellence on Lymph Vessels in Obesity and Cardiovascular Disease (11CVD03), the Synapsis Foundation and the Heidi Seiler-Stiftung.

## Author contributions

Q.M., B.V.I., M.D. and S.T.P. conceived and designed the study; Q.M., B.V.I and S.T.P. performed the experiments and analyzed the data; and Q.M., B.V.I., M.D. and S.T.P. drafted the manuscript. All authors have approved the final version of the manuscript and have agreed to be accountable for all aspects of the work.

## Additional information

**Competing interests:** The authors declare no competing financial interests.

