## [Peer Review File · Nature Communications]

Reviewers' comments:

Reviewer #1 (Remarks to the Author):

Review for Nature Communications of manuscript NCOMMS-17-00229 entitled "Lymphatic outflow of cerebrospinal fluid is reduced in aged mice"

By Qiaoli Ma, Michael Detmar and Steven T. Proulx

General comments:

1. The authors of this study use state-of-the-art methods to identify new features of the flow of cerebrospinal fluid (CSF) from the lateral ventricles of the mouse brain into lymphatic vessels that drain the head and connect to the bloodstream. Strengths of the work include the use of novel tracers, Prox1-GFP lymphatic reporter mice, fluorescence imaging, and quantitative data for the rate of accumulation in the bloodstream of tracers infused into a brain ventricle.

2. Of interest is the use of innovative methods and fluorescent P40D680, a novel pegylated near-infrared 40 kDa tracer, to obtain quantitative data on the rate of accumulation in the bloodstream of the saphenous vein after infusion into CSF of a lateral ventricle. Other novel findings include the comparison of accumulation rates in cervical lymph nodes and the bloodstream and the comparison of P40D680 to Evans blue. Importantly, the authors also report that the tracer did not enter dural lymphatics, which have previously been interpreted as routes for CSF drainage.

3. Despite the many attributes, shortcomings involving many parts of the manuscript weaken the story and make some of the claims difficult to accept. These are described in the following comments.

4. More balanced historical perspective needed: Background information describing what is known and what is not known about CSF production and clearance and the relevant CNS compartments and barriers involved is oversimplified and skewed.

a. The first sentence of the Introduction, which states that CSF "is considered to be produced...by the choroid plexus", ignores the compelling literature on non-choroid plexus sources of CSF production, as demonstrated by choroid plexus removal and other approaches. The description deserves better balance.

b. The pattern and routes of CSF flow and clearance also deserve more balanced consideration. Although the authors' mention the known clearance of CSF through lymphatics associated with cranial nerves, the description does not adequately acknowledge the extensive literature on this topic, including more than 30 papers by Miles Johnston and colleagues alone, and seems to trivialize how effectively lymphatic clearance of CSF has been documented by studies of many species including mice. As a consequence, despite the innovative methods, it is unclear which findings reported are new and which are confirmatory. A more representative treatment of the background is essential for readers to understand which data are novel and which confirm earlier work.

c. Please acknowledge that CSF outflow via lymphatics is known to occur in the sheath of spinal nerves as well as in the sheath of cranial nerves. Mention should also be made of evidence of clearance of CSF through spinal veins. In this regard, clarification is needed to distinguish segments of nerves within the subarachnoid space (central nervous system) from more distal parts of the same nerves covered by epineurium (peripheral nervous system).

d. CSF circulation pathways and the meningeal barriers involved should also be more clearly described

in the context of the findings. The use of "meningeal lining" obfuscates the sheath(s) involved. Please explain that, before they penetrate the brain, major arteries, e.g., arteries of the circle of Willis, are in the subarachnoid space where CSF circulates. Also emphasize that the arachnoidal barrier isolates the subarachnoid space from the dura and thereby separates the CSF from lymphatics in the dura. This issue is key to understanding the access and function of lymphatics within the dura.

e. Comments in the Discussion (page 16) about tight junctions do not capture the concept of barriers that isolate the brain and spinal cord. Please revise this section after further review of the relevant literature, e.g., Brightman MW, Reese TS. *J Cell Biol.* 40(3): 648-77, 1969; Brightman MW et al. *J Neurol Sci.* 10(3): 215-39, 1970; Reese TS et al. *J Neuropathol Exp Neurol.* 30(1): 137-8, 1971; Nabeshima S et al. *J Comp Neurol.* 164(2): 127-69, 1975; Begley DJ, Brightman MW. *Prog Drug Res.* 61: 39-78, 2003.

f. Also please describe that the same barriers apply to connective tissue sheaths of extracranial nerves, where endoneurium, perineurium, and epineurium have properties equivalent to pia, arachnoid, and dura, respectively. In both, the barrier is at the level of the arachnoid or perineurium. The authors should more clearly define the region of cranial nerves examined (intracranial or extracranial) and the sheath (endoneurium, perineurium, or epineurium) in which lymphatics were located.

g. The authors' skepticism over arachnoidal villi as sites of CSF outflow is not adequately balanced by evidence supporting this route of CSF clearance. The authors should acknowledge that this has been a topic of dozens if not hundreds of papers, many of which have led to the conclusion that CSF exits the subarachnoid space through arachnoidal villi, lymphatics, and other routes.

5. Better definition of questions addressed: After consideration of the issues described in Comment 4, the authors should more clearly define in the Introduction the specific questions they sought to answer and help the reader understand why their innovative methodology would be successful in answering these questions where others failed previously. In other words, explain what new information could be obtained and what puzzling questions could be answered by the new approaches.

6. No CSF clearance via arachnoidal villi: The interpretation (Results, page 10, lines 247-249) that the dynamic imaging data "indicates" that the tracer measured in blood reached the circulation through lymphatics and NOT via arachnoidal villi is unconvincing. The absence of concurrent measurements of tracer accumulation in the superior sagittal sinus or other collecting veins precludes the exclusion of this pathway. The argument that the delayed appearance of tracer in the bloodstream means that lymphatics - but not arachnoid villi - were involved does not make sense. What is the basis of the assumption that transport into blood via arachnoidal villi is faster than via lymphatics? In any case, this interpretation belongs in the Discussion, not in the Results, and the claim should be accompanied by consideration of the strengths and limitations of the methods and alternative interpretations.

7. Technical factors that could limit detection of tracer in dural lymphatics: Absence of the CSF tracer in dural lymphatics is a novel observation that has important implications but could be influenced by technical issues that are not addressed. Images in Supplemental Figure 4 that support the claim seem anecdotal and not very convincing. The threshold of detection of P40D680 under these conditions is unclear. Does background fluorescence of the dura mask P40D680 fluorescence? The low resolution of dissecting microscopic images is another limiting factor. In addition, the time course of tracer accumulation, which matches measurements in the saphenous vein, should be included, as should the number of mice examined at each time point.

8. Age-related changes in CSF outflow: Claims of novelty of slower CSF tracer entry into the

bloodstream in older mice needs a more balanced historical perspective and more critical consideration of the underlying mechanisms. The authors should acknowledge what is known about decreased CSF production with aging, restricted CSF outflow with aging, changes in the choroid plexus and arachnoidal villi with aging, and related topics. Also of importance is published evidence for increased CSF volume with aging, which could contribute to the observed results by augmenting the dilution of tracer infused into a larger volume of CSF in enlarged ventricles of older mice.

9. CSF pressure during infusion: Did infusion of tracer at 1 μ l/min increase CSF pressure? The authors' argue that changes in CSF pressure can alter CSF clearance pathways and mechanisms. Measurements of intraventricular pressure before and during infusion of tracer would strengthen the results by documenting the conditions under which the observations of lymphatic clearance were made.

10. Evans blue:

a. The use of a low molecular weight fluorophore for comparison to 40 kDa P40D680 makes sense, but the behavior of Evans blue (960 Da) as a tracer, including binding to albumin and to cellular elements of brain, is strongly influenced by the electronegative surface potential and other properties that differ from those of fluorescein and indocyanine green.

b. The authors should acknowledge that the pattern of Evans blue shown in Figure 5 and described on pages 11-12 is likely to be determined in part by the selective binding due to the molecular properties of Evans blue.

c. This part of the manuscript would be strengthened by addition of corresponding data for a tracer of similar size that does not have the binding properties of Evans blue.

d. Description of Evans blue binding to the adventitia of arteries (Figure 5) is not supported by convincing evidence that this is the actual site of binding. Please add colocalization data that make it possible to determine whether the dye binds to components of the adventitia, medial smooth muscle, or endothelium.

e. Acknowledge that little or no Evans blue staining of veins could reflect less binding to the abluminal surface of this segment of the vasculature.

11. Figures:

a. Some of the figures are difficult to interpret and would benefit from enlargement, higher resolution confocal microscopic images, and accompanying diagrams for orientation. Clearly labeled diagrams are recommended especially for Figures 2a-e, 4a, c, and Supplemental Figure 2a-d, 4a-f.

b. The orientation diagrams of Supplemental Figure 3a-b are sufficiently important to warrant moving them into the main manuscript.

c. Supplemental Figure 3b should be revised to more closely match mouse anatomy.

d. Figure panels in the Supplemental Figures should be enlarged to improve visibility by taking advantage of the entire page.

e. Supplemental Figure 7 would be easier to understand as a schematic diagram if it more closely matched mouse anatomy. Please correct points of ambiguity: The combined 2-dimensional/3-dimensional perspective is difficult to interpret; mandibular lymph nodes appear to be in the shoulder; deep cervical lymph nodes appear to be in the foreleg; the location of meningeal lymphatics is unclear

(also see Comment 4d); the location of cranial nerves does not fit with anatomy; 3 optic nerves are shown on the right side; vibrissae are shown larger than cranial nerves and lymphatics.

f. The Supplemental videos are not very informative because of the absence of labels and inadequate description in the legends of the purpose and what viewers should look for in each video.

12. Critical consideration of results in Discussion: The Discussion reads more like a review than a critical assessment of the authors' methods and results. Readers would benefit from an initial summary of the main new findings (only results, not interpretations) followed by discussion the strengths and limitations of the methods and the findings. The Discussion should include evidence for and against CSF clearance via arachnoidal villi, potential significance and pitfalls of not finding tracer in dural lymphatics, and the historical context of age-related changes in CSF outflow.

Specific comments:

1. Description of Evans blue binding to the "adventitia" of arteries should be accompanied by more compelling evidence that this is actually the site of binding. Please explain how, in these low magnification images, the staining of adventitia was distinguished from staining of the media and endothelium. Higher magnification confocal microscopic images showing co-staining of smooth muscle cells, endothelial cells, and basement membranes should help with the clarification.

2. Please define in physical terms what is meant by "counts" in fluorescence imaging signal enhancement. Counts of what? Photons? What is the denominator? Time? Area?

3. "Close proximity": Does this mean something different from "proximity"?

4. "Bolus infusion" seems to be an oxymoron. As 2.5 μ l of tracer was given over 2.5 minutes, this would seem to be an infusion instead of a bolus injection (a single dose given all at once).

5. "intraventricle infusion" should be "intraventricular infusion"

6. The number of mice studied should be reported for all experiments. Please give the N (number of mice) for all quantitative data, including Figures 3b, 3d, 4d. Also provide the number and age of mice examined in imaging studies of lymphatic drainage and dural lymphatics.

7. Comments about the absence of tracer along the "olfactory nerves" (page 6, line 149) should be clarified by describing exactly where the observations were made. Were olfactory nerves distinguished from the olfactory bulb and olfactory tract? Were the olfactory bulb and tract stained by tracer in the subarachnoid space? Were olfactory nerves examined as they pass through the cribriform plate, olfactory mucosa, or both? The corresponding information should be given for other cranial nerves examined.

8. Citation in the Discussion (page 16, line 372) of reference 41, which was published in 1957, to document the presence of tight junctions in CNS blood vessels is invalid because tight junctions were not described as the basis of the barrier function of CNS blood vessels for another twelve years (J Cell Biol 40(3): 648-77, 1969).

Reviewer #2 (Remarks to the Author):

This is an ex vivo and in vivo study seeking to define lymphatic CSF outflow pathways from the cranium, and to define the effect of aging on the lymphatic clearance of CSF. The authors provide a detailed description of different peri-neural and other pathways by which CSF tracer exits the cranium, including providing some data that is at odds with findings from two recent papers^{1,2} describing tracer uptake into lymphatic vessels associated with the dural sinuses. Using both ex vivo and in vivo imaging of CSF tracer along lymphatic pathways, the authors provide evidence that lymphatic clearance of CSF is slowed in the aging versus the young adult brain. This is a well conceived and well written study, that makes an important contribution to the present understanding of lymphatic cerebrospinal fluid transport in the context of aging which may have important implications in the pathophysiology of neurodegenerative disease.

However, in several instances the conclusions drawn by the authors are somewhat overstated, or perhaps go beyond where the data that they provide would reasonably permit. These should be addressed, and are outlined in the comments below.

Major Comments

1. In the results section, the authors state "The delay in the time for the signal to be apparent in the blood indicated that there did not appear to be direct venous uptake of the tracer, implying that arachnoid villi or other possible direct routes into the blood were not active under these conditions." Throughout the study, the authors forcefully make the assertion that the slow appearance of CSF tracer into the saphenous blood pool indicates the absence of direct CSF-venous blood communication. An alternative interpretation of these data is that the delay of detectable tracer in the venous blood is due to dilution of the intravascular tracer below the authors' detection limit. The authors' conclusion regarding direct vascular uptake of the tracer should reflect this possibility, unless they can provide more direct evidence as to the lack of involvement of direct CSF-venous blood communication. Similarly, the editorial comments within the Results section related to these findings "...implying that arachnoid villi or other possible direct routes into the blood were not active...", "In summary, the results of the dynamic imaging indicate that in mice the bulk of the outflow from the CSF of macromolecular and small molecular tracers was transported through the lymphatic system rather than venous routes", etc. should be removed, and restricted to a more balanced treatment of this important issue in the Discussion.

2. The notion that lymphatic clearance of CSF is slowed in the aging brain is important, and has many critical implications for our understanding of age-related pathology, including neurodegenerative diseases; thus the observation that CSF tracer injected into the lateral ventricle appears more slowly in the lymphatic drainage and saphenous blood pool is potentially very important. However, the movement of CSF tracer through the ventricular system to the blood pools is not dependent solely upon lymphatic clearance. It is also presumably influenced by the rate of CSF secretion, and bulk/dispersionary transport along the ventricular compartment. Given that CSF secretion is known to be reduced in the aging brain^{3,4}, it must be considered that the delayed appearance in the blood pool reflects the slowed transit through the ventricular system, versus the slowed clearance along the lymphatic vasculature. One way to address this would be to repeat these experiments using injection into a cisternal CSF compartment, thus bypassing the ventricular system.

3. In their discussion, the authors state "Our study is, to our knowledge, the first to characterize in detail the lymphatic outflow pathways of CSF from the skull in mice (Supplemental Fig. 7). Our use of pegylated NIR tracers and lymphatic-specific reporter mice combined with high-resolution stereomicroscopy have allowed an in-depth analysis that was not previously possible." The use of a

pegylated IR dye tracer in a lymphatic-specific reporter was previously employed by Aspleund et al., 2015 (DOI: 10.1084/jem.20142290) for exactly this purpose. Further, detailed pathways of lymphatic efflux from the central nervous system can be found in the literature throughout the last forty years, including many citations included within this manuscript. Therefore, the multiple statements throughout this manuscript, highlighting the novelty of these findings should be moderated.

4. Within the introduction, the authors state, "Next, we used a recently developed tracer transport to blood assay to evaluate the dynamics of CSF outflow to the systemic circulation of a pegylated NIR dye macromolecular tracer in comparison to Evans blue, which behaves as a small molecule within the low protein CSF." Evans blue avidly associates with albumin as discussed in the body of this manuscript, thus depending on the saturation of EB with albumin, a fraction of this tracer would behave more as a ~70 kDa protein, not a small molecule. Even if the presence of albumin in the CSF is lower than that of the blood, the 2.5 μ L injected may still be saturated. Do the authors know the fraction of the injected EB that is free vs. bound? If the authors wish to model the movement of a small molecule through the ventricular system and subarachnoid spaces, using an inert tracer such as a fluorescent dextran seemingly would be more appropriate.

Minor comments on figures

4d- Given that the authors collected dynamic imaging of bilateral deep cervical lymph nodes, it is unclear why the authors do not report mean enhancement +/- SD as in 4b.

4e- Please annotate this panel with the subset of lymph nodes under consideration.

1. Aspelund A, Antila S, Proulx ST, et al. A dural lymphatic vascular system that drains brain interstitial fluid and macromolecules. *J Exp Med*. 2015;212(7):991-999.
2. Louveau A, Smirnov I, Keyes TJ, et al. Structural and functional features of central nervous system lymphatic vessels. *Nature*. 2015;523(7560):337-341.
3. Stoquart-EISankari S, Baledent O, Gondry-Jouet C, Makki M, Godefroy O, Meyer ME. Aging effects on cerebral blood and cerebrospinal fluid flows. *J Cereb Blood Flow Metab*. 2007;27(9):1563-1572.
4. May C, Kaye JA, Atack JR, Schapiro MB, Friedland RP, Rapoport SI. Cerebrospinal fluid production is reduced in healthy aging. *Neurology*. 1990;40(3 Pt 1):500-503.

Response to reviewer comments:

Reviewer #1:

General comments:

1. *The authors of this study use state-of-the-art methods to identify new features of the flow of cerebrospinal fluid (CSF) from the lateral ventricles of the mouse brain into lymphatic vessels that drain the head and connect to the bloodstream. Strengths of the work include the use of novel tracers, Prox1-GFP lymphatic reporter mice, fluorescence imaging, and quantitative data for the rate of accumulation in the bloodstream of tracers infused into a brain ventricle.*

2. *Of interest is the use of innovative methods and fluorescent P40D680, a novel pegylated near-infrared 40 kDa tracer, to obtain quantitative data on the rate of accumulation in the bloodstream of the saphenous vein after infusion into CSF of a lateral ventricle. Other novel findings include the comparison of accumulation rates in cervical lymph nodes and the bloodstream and the comparison of P40D680 to Evans blue. Importantly, the authors also report that the tracer did not enter dural lymphatics, which have previously been interpreted as routes for CSF drainage.*

3. *Despite the many attributes, shortcomings involving many parts of the manuscript weaken the story and make some of the claims difficult to accept. These are described in the following comments.*

Response: We thank the reviewer for the kind comments.

4. *More balanced historical perspective needed: Background information describing what is known and what is not known about CSF production and clearance and the relevant CNS compartments and barriers involved is oversimplified and skewed.*

Response: Since we are limited by word counts, we are forced to simplify the introduction and discussion of the background information on CSF production and clearance, as well as the barriers involved. We have tried best we can to address the following comments without increasing the word count dramatically.

a. *The first sentence of the Introduction, which states that CSF “is considered to be produced...by the choroid plexus”, ignores the compelling literature on non-choroid plexus sources of CSF production, as demonstrated by choroid plexus removal and other approaches. The description deserves better balance.*

Response: We agree with the reviewer that specification of the sole source of CSF as the choroid plexus is a limited description. We have now revised the introduction to state that

extrachoroidal plexus sources of CSF fluid exist, namely in the form of interstitial fluid likely derived from the blood-brain barrier.

b. The pattern and routes of CSF flow and clearance also deserve more balanced consideration. Although the authors' mention the known clearance of CSF through lymphatics associated with cranial nerves, the description does not adequately acknowledge the extensive literature on this topic, including more than 30 papers by Miles Johnston and colleagues alone, and seems to trivialize how effectively lymphatic clearance of CSF has been documented by studies of many species including mice. As a consequence, despite the innovative methods, it is unclear which findings reported are new and which are confirmatory. A more representative treatment of the background is essential for readers to understand which data are novel and which confirm earlier work.

Response: We have now expanded our description of the lymphatic outflow pathways in the introduction, including what is known about the perineural outflow routes. The work of Miles Johnston and colleagues, which was a big influence on the present study, was especially mentioned in the Discussion and cited repeatedly. We are limited to 70 citations so unfortunately not all of the previous literature could be cited despite its merit. However, we were careful to ensure that all of the major groups that have investigated this topic in the last 70 years were cited, namely Brierley and Field (2 cites), Földi (2), Arnold (2), Bradbury (3), Cserr (2), McComb (7), Weller (1), Brinker (1) and Johnston (8).

c. Please acknowledge that CSF outflow via lymphatics is known to occur in the sheath of spinal nerves as well as in the sheath of cranial nerves. Mention should also be made of evidence of clearance of CSF through spinal veins. In this regard, clarification is needed to distinguish segments of nerves within the subarachnoid space (central nervous system) from more distal parts of the same nerves covered by epineurium (peripheral nervous system).

Response: We have now acknowledged in the introduction that spinal outflow pathways to lymphatics via the sheath of spinal nerves are known. We also cite evidence for spinal arachnoid villi. While we have also observed evidence of spinal outflow to lymphatic vessels, this was not the focus of the current manuscript and will be described as part of a future report. Consistent with previous reports this outflow appeared minor under normal conditions compared to the cranial outflow pathways. We are not proposing flow along the peripheral nerves as it appears that the tracers are reaching lymphatic vessels just outside the skull in agreement with previous reports.

d. CSF circulation pathways and the meningeal barriers involved should also be more clearly described in the context of the findings. The use of "meningeal lining" obfuscates the sheath(s) involved. Please explain that, before they penetrate the brain, major arteries, e.g., arteries of the circle of Willis, are in the subarachnoid space where CSF circulates. Also emphasize that the arachnoidal barrier isolates the subarachnoid space from the dura and thereby separates the CSF from

lymphatics in the dura. This issue is key to understanding the access and function of lymphatics within the dura.

Response: We have removed mention of the generic term “meningeal lining” and have included a description of the meningeal layers surrounding the subarachnoidal space (SAS) and the dura mater in the Introduction: “Two meningeal layers, the leptomeninges, surrounding the SAS restrict flow of CSF into the brain parenchyma (pia mater, that associates with glial cells to form the semi-permeable pial-glial membrane) and prevent access to the outermost meningeal layer, the dura mater (the tight junction-containing arachnoid layer)” We have also now included mention that the arteries of the circle of Willis are within the SAS. Description of the arachnoid barrier was previously only in the Discussion, mention of this has also now also been added to the Introduction when discussing the dural mater lymphatics: “However, this potential route for drainage of CSF is controversial in light of the existence of the arachnoid barrier layer between the SAS and the dura mater (Nabeshima et al, 1975).”

e. Comments in the Discussion (page 16) about tight junctions do not capture the concept of barriers that isolate the brain and spinal cord. Please revise this section after further review of the relevant literature, e.g., Brightman MW, Reese TS. J Cell Biol. 40(3): 648-77, 1969; Brightman MW et al. J Neurol Sci. 10(3): 215-39, 1970; Reese TS et al. J Neuropathol Exp Neurol. 30(1): 137-8, 1971; Nabeshima S et al. J Comp Neurol. 164(2): 127-69, 1975; Begley DJ, Brightman MW. Prog Drug Res. 61: 39-78, 2003.

Response: We thank the reviewer for indicating the relevant literature regarding the anatomy of the barriers of the CNS. After review of these papers we have added a sentence to the Discussion: “Similar to the barriers at the endothelium of the brain capillaries and at the epithelial lining of the choroid plexus (Brightman and Reese, 1969), there exists a barrier layer that consists of arachnoid cells containing tight junctions that serves to isolate the CSF within the SAS from the interstitial fluid of the dura mater (Nabeshima et al, 1975).” Both of these papers have now been cited in the Introduction and Discussion (Refs 3 and 4).

f. Also please describe that the same barriers apply to connective tissue sheaths of extracranial nerves, where endoneurium, perineurium, and epineurium have properties equivalent to pia, arachnoid, and dura, respectively. In both, the barrier is at the level of the arachnoid or perineurium. The authors should more clearly define the region of cranial nerves examined (intracranial or extracranial) and the sheath (endoneurium, perineurium, or epineurium) in which lymphatics were located.

Response: Our data is consistent with many previous reports describing a perineural outflow route along intracranial nerves to just outside the skull, within a sheath that is continuous with the SAS. We have now cited these reports in the introduction and discussion (Refs. 20-27). We are not proposing flow along the peripheral nerves, nor the existence of lymphatics within the nerve sheaths themselves, therefore, we have not

expanded our discussion to include the barriers in the peripheral nerve sheaths.

g. The authors' skepticism over arachnoidal villi as sites of CSF outflow is not adequately balanced by evidence supporting this route of CSF clearance. The authors should acknowledge that this has been a topic of dozens if not hundreds of papers, many of which have led to the conclusion that CSF exits the subarachnoid space through arachnoidal villi, lymphatics, and other routes.

Response: We respectfully disagree with the reviewer regarding the existence of strong physiological evidence supporting the role of arachnoid villi or granulations as major sites of CSF outflow. The reviewer is correct that there is an extensive body of literature on these structures, going back to the seminal studies of Key and Retzius (1876) and Lewis Weed (1914) that established the traditional textbook understanding of CSF outflow. Several early studies did demonstrate rapid outflow of dyes to blood and appear to have influenced the conclusions of Weed, but as Goldmann (Vitalfärbung am Zentralnervensystem, 1913) suggested, these studies did not adequately control for injection pressure. We have searched in vain for published in vivo data that indicate a direct uptake into blood of a bulk flow tracer after injection into the CSF compartment under controlled pressure conditions. The strongest evidence for such a route is Davson et al., *Brain*, 1973 in which radiolabeled albumin was continuously infused into ventricles of the rabbit and blood signal was tested after various time points after the start of the infusion. However, the earliest time point tested was 30 min which would not rule out a lymphatic route. In fact, one could argue that stronger evidence exists in the literature against a direct blood uptake with several studies (Refs. 13, 61-65 as cited in the Discussion) in which cannulated blood from either within the superior sagittal sinus or draining veins did not exhibit higher concentrations of tracers compared to arterial samples. Nonetheless, we have expanded our discussion of arachnoid villi and granulations in the introduction as well as established the concept of a dual outflow system as proposed by Pollay, 2010 (Ref. 7).

5. Better definition of questions addressed: After consideration of the issues described in Comment 4, the authors should more clearly define in the Introduction the specific questions they sought to answer and help the reader understand why their innovative methodology would be successful in answering these questions where others failed previously. In other words, explain what new information could be obtained and what puzzling questions could be answered by the new approaches.

Response: We have revised the last two paragraphs of the introduction to define the specific questions that the paper seeks to clarify.

6. No CSF clearance via arachnoidal villi: The interpretation (Results, page 10, lines 247-249) that the dynamic imaging data "indicates" that the tracer measured in blood reached the circulation through lymphatics and NOT via arachnoidal villi is unconvincing. The absence of concurrent measurements of tracer accumulation in the superior sagittal sinus or other collecting veins

precludes the exclusion of this pathway. The argument that the delayed appearance of tracer in the bloodstream means that lymphatics - but not arachnoid villi - were involved does not make sense. What is the basis of the assumption that transport into blood via arachnoidal villi is faster than via lymphatics? In any case, this interpretation belongs in the Discussion, not in the Results, and the claim should be accompanied by consideration of the strengths and limitations of the methods and alternative interpretations.

Response: We agree with the reviewer that simply interpreting the dynamic data of a delay of uptake into the blood does not rule out a direct blood pathway. The reviewer is correct that our initial interpretation assumed that a transport of tracer through arachnoid villi would be a more rapid process than lymphatic outflow. However, this assumption of rapid outflow through villi was shared by many previous investigators in this field (e.g. Weed, 1914, Welch and Friedman, *Brain*, 1960, Davson et al, *Brain*, 1973). It is true, however, that since the mechanism for transport through the villi has never been determined to be through an open (e.g. one-way valve) or closed (e.g. pinocytosis through an endothelial cell barrier) system, it is impossible to predict how fast of a process this would be. Therefore, in a further attempt to rule out direct venous outflow we imaged the posterior facial vein after infusion into the lateral ventricle (in n=5 mice). These veins on each side of the mouse collect the blood from the transverse sinus that exits at the postglenoid foramen. At no point were we able to detect an increase in signal at this vein until it was obvious that a systemic blood increase had occurred. At the termination of the experiment 30 min after infusion, there was bright signal in the collecting lymphatics tracking towards both the mandibular and deep cervical lymph nodes, as well as an obvious increase in signal at the saphenous vein (new Supplemental Figure 5). Also, in our investigations of the deep cervical region at 10 and 30 min there was never any apparent signal within the jugular veins. However, we are unable to eliminate a scenario in which a direct blood outflow occurs at later time points. This possibility is now discussed. Throughout the manuscript we have been careful to state that blood outflow cannot be absolutely be ruled out and at no point do we make the claim that all outflow is lymphatic in nature.

7. Technical factors that could limit detection of tracer in dural lymphatics: Absence of the CSF tracer in dural lymphatics is a novel observation that has important implications but could be influenced by technical issues that are not addressed. Images in Supplemental Figure 4 that support the claim seem anecdotal and not very convincing. The threshold of detection of P40D680 under these conditions is unclear. Does background fluorescence of the dura mask P40D680 fluorescence? The low resolution of dissecting microscopic images is another limiting factor. In addition, the time course of tracer accumulation, which matches measurements in the saphenous vein, should be included, as should the number of mice examined at each time point.

Response: These images are collected under similar settings used to easily visualize tracer within the lymphatic vessels outside the skull, as well as in perineural areas at the base of the skull. Background autofluorescence signals are very weak in the near-infrared

wavelengths, so we don't believe that this would mask tracer signal and prevent us from detecting tracer within the Prox1-GFP⁺ lymphatic vessels. Of course, as with all imaging techniques, a detection limit exists and there is a possibility that some small amount of tracer has reached these lymphatic vessels. However, in light of the known anatomical barriers and the much brighter perineural outflow signals, we feel confident in stating that the dural lymphatic vessels in the regions examined do not appear to be a large contributor to CSF outflow under normal conditions in mice.

8. Age-related changes in CSF outflow: Claims of novelty of slower CSF tracer entry into the bloodstream in older mice needs a more balanced historical perspective and more critical consideration of the underlying mechanisms. The authors should acknowledge what is known about decreased CSF production with aging, restricted CSF outflow with aging, changes in the choroid plexus and arachnoidal villi with aging, and related topics. Also of importance is published evidence for increased CSF volume with aging, which could contribute to the observed results by augmenting the dilution of tracer infused into a larger volume of CSF in enlarged ventricles of older mice.

Response: We have now acknowledged previous knowledge regarding decreased CSF outflow of tracers and proteins in aged rats, as well as aged-related decreases in CSF production and changes in the structure of the choroid plexus as outlined in the review by Preston, *Microsc Res Tech*, 2001. We have also indicated the published evidence for increased CSF volume in aged humans and rats and acknowledged that an increased volume of CSF could dilute the infused tracer.

9. CSF pressure during infusion: Did infusion of tracer at 1 μ l/min increase CSF pressure? The authors' argue that changes in CSF pressure can alter CSF clearance pathways and mechanisms. Measurements of intraventricular pressure before and during infusion of tracer would strengthen the results by documenting the conditions under which the observations of lymphatic clearance were made.

Response: This is a valid critique and one that we are unable to fully address due to the lack of a setup to measure intracranial pressures in mice. It is possible that a temporary increase in intraventricular pressure occurs during the infusion that may accelerate flow through the ventricular system, but we believe as shown in previous reports after cisterna magna injections (e.g. Ref. 38) that CSF pressures will quickly normalize. We did, however, attempt to address the reviewer's comment by infusing tracers at lower volumes (1 μ L of 500 μ M P40D680 over 2.5 min; n=3 mice) and slower rates (2.5 μ L of 200 μ M P40D680 over 5 min; n=3 mice) into the lateral ventricles. The anatomical patterns of perineural outflow were unchanged. As shown here for the reviewer, there was a substantial increase in the transit time to blood with the 500 μ M P40D680 infusion, however, it is not clear whether this effect was due to a reduction in intraventricular pressure or an increased viscosity of the tracer, which has been shown previously to flow more slowly through the ventricles (Davson et al, *Brain*, 1970). Our results after cisterna magna infusions, showed similar transport to blood dynamics (new Supplemental Figure 9) and anatomical routing (not shown), so we believe that we have not substantially

altered CSF clearance pathways or mechanisms with our ventricular infusion protocol.

10. *Evans blue*:

a. The use of a low molecular weight fluorophore for comparison to 40 kDa P40D680 makes sense, but the behavior of *Evans blue* (960 Da) as a tracer, including binding to albumin and to cellular elements of brain, is strongly influenced by the electronegative surface potential and other properties that differ from those of fluorescein and indocyanine green.

b. The authors should acknowledge that the pattern of *Evans blue* shown in Figure 5 and described on pages 11-12 is likely to be determined in part by the selective binding due to the molecular properties of *Evans blue*.

c. This part of the manuscript would be strengthened by addition of corresponding data for a tracer of similar size that does not have the binding properties of *Evans blue*.

Response: We fully agree that *Evans blue* has many limitations as a tracer (as nicely summarized in Saunders et al., *Frontiers Neuro*, 2015) and only chose this dye due to its ability to bind albumin once it reached the blood, thereby increasing its serum-half life and detectability in systemic blood. After demonstrating that it would be possible to detect with high sensitivity other small molecular tracers at the saphenous vein region after i.v. injections (new Supplemental Figure 6), we performed studies (new Figure 6 and new Supplemental Figure 7) with intraventricular infusions of two additional small molecular tracers, IRDye680 CW (which does not bind albumin but still exhibits binding to the arteries) and AlexaFluor680-3kDa dextran (which does not bind albumin nor bind substantially to arteries, but exhibits cellular uptake). We were unable to identify a small molecular tracer similar to P40D680 that does not show at least some retention in the brain. Regardless, as shown in the new Figure 6, the data from all three small molecules

indicated that there was outflow from the CSF with a delayed uptake to systemic blood and signal within CSF-draining lymphatic vessels.

d. Description of Evans blue binding to the adventitia of arteries (Figure 5) is not supported by convincing evidence that this is the actual site of binding. Please add colocalization data that make it possible to determine whether the dye binds to components of the adventitia, medial smooth muscle, or endothelium.

e. Acknowledge that little or no Evans blue staining of veins could reflect less binding to the abluminal surface of this segment of the vasculature.

Response: With the addition of the data from two other small molecular tracers, we did not believe it was critical to report the exact binding location of Evans blue to arteries. Therefore, we have removed mention of adventitial binding in the manuscript. The sentence now reads “Fluorescence imaging indicated more significant accumulation of EB and IRDye680CW around arteries compared to veins (new Supplemental Figure 7b), rather than the paravascular spreading pattern that was observed around both blood vessel types with P40D680.”

11. Figures:

a. Some of the figures are difficult to interpret and would benefit from enlargement, higher resolution confocal microscopic images, and accompanying diagrams for orientation. Clearly labeled diagrams are recommended especially for Figures 2a-e, 4a, c, and Supplemental Figure 2a-d, 4a-f.

Response: We have separated Figure 2 into two figures (new Figures 3 and 4) to allow enlargement of figure panels along with orientation diagrams to indicate where each image was acquired to document the outflow routes. We have also provided labeling for the dural lymphatic vessel images in the new Supplemental Figure 3. We refer to the new Figure 3d in the legend of new Figure 5c (previous Figure 4c) to indicate the region that the videos over the mandibular lymph nodes are acquired.

b. The orientation diagrams of Supplemental Figure 3a-b are sufficiently important to warrant moving them into the main manuscript.

Response: We have moved Supplemental Figure 3a and 3b into the main manuscript (as new Figures 3d and 4d). We have also added an orientation diagram for the base of the skull (new Figure 3c).

c. Supplemental Figure 3b should be revised to more closely match mouse anatomy.

Response: Supplemental Figure 3b (new Figure 4d) has been revised to include more anatomical features of the mouse.

d. Figure panels in the Supplemental Figures should be enlarged to improve

visibility by taking advantage of the entire page.

Response: We have enlarged the Supplemental Figure panels as much as possible.

e. Supplemental Figure 7 would be easier to understand as a schematic diagram if it more closely matched mouse anatomy. Please correct points of ambiguity: The combined 2-dimensional/3-dimensional perspective is difficult to interpret; mandibular lymph nodes appear to be in the shoulder; deep cervical lymph nodes appear to be in the foreleg; the location of meningeal lymphatics is unclear (also see Comment 4d); the location of cranial nerves does not fit with anatomy; 3 optic nerves are shown on the right side; vibrissae are shown larger than cranial nerves and lymphatics.

Response: This figure (new Supplemental Figure 10) has been revised to more closely match the anatomical features of the mouse.

f. The Supplemental videos are not very informative because of the absence of labels and inadequate description in the legends of the purpose and what viewers should look for in each video.

Response: We have added title slides to the videos and expanded the descriptions in the video legends.

12. Critical consideration of results in Discussion: The Discussion reads more like a review than a critical assessment of the authors' methods and results. Readers would benefit from an initial summary of the main new findings (only results, not interpretations) followed by discussion the strengths and limitations of the methods and the findings. The Discussion should include evidence for and against CSF clearance via arachnoidal villi, potential significance and pitfalls of not finding tracer in dural lymphatics, and the historical context of age-related changes in CSF outflow.

Response: We have revised the discussion section to incorporate the reviewer's suggestions.

Specific comments:

1. Description of Evans blue binding to the "adventitia" of arteries should be accompanied by more compelling evidence that this is actually the site of binding. Please explain how, in these low magnification images, the staining of adventitia was distinguished from staining of the media and endothelium. Higher magnification confocal microscopic images showing co-staining of smooth muscle cells, endothelial cells, and basement membranes should help with the clarification.

Response: As mentioned above, we have removed mention of binding of EB to adventitia

of the arteries, as this information is not critical to the conclusions of the paper.

2. *Please define in physical terms what is meant by “counts” in fluorescence imaging signal enhancement. Counts of what? Photons? What is the denominator? Time? Area?*

Response: Counts is a commonly used unit that is generated by CCD cameras as a digital measure of signal based on the number of photons that reach the imaging sensor during the exposed time. It requires an analog-to-digital conversion from photons to electrons. The measure would theoretically be expressed as a unit of both area (pixel size) and time, but as these parameters can vary widely with magnification, binning of pixels and the exposure time during image acquisition, the raw value in counts as an arbitrary unit is usually expressed. As long as comparisons are made between images acquired using the same settings then it is valid to use this as a quantitative unit. Our data showing direct relationships between known blood concentrations and fluorescence signal is direct proof of the quantitative nature of such an approach.

3. *“Close proximity”: Does this mean something different from “proximity”?*

Response: We concur that this term could seem redundant, but it is perfectly valid English. Merriam-Webster’s Dictionary of English Usage says, “Of course there are degrees of proximity, and close proximity simply emphasizes the closeness.”

4. *“Bolus infusion” seems to be an oxymoron. As 2.5 μ l of tracer was given over 2.5 minutes, this would seem to be an infusion instead of a bolus injection (a single dose given all at once).*

Response: We agree and have changed all examples in the text to simply “infusion”.

5. *“intraventricle infusion” should be “intraventricular infusion”*

Response: We agree and have corrected all terms.

6. *The number of mice studied should be reported for all experiments. Please give the N (number of mice) for all quantitative data, including Figures 3b, 3d, 4d. Also provide the number and age of mice examined in imaging studies of lymphatic drainage and dural lymphatics.*

Response: This information has been added to all figure legends.

7. *Comments about the absence of tracer along the “olfactory nerves” (page 6, line 149) should be clarified by describing exactly where the observations were made. Were olfactory nerves distinguished from the olfactory bulb and olfactory tract? Were the olfactory bulb and tract stained by tracer in the subarachnoid space? Were olfactory nerves examined as they pass through the cribriform plate, olfactory mucosa, or both? The corresponding information should be given for*

other cranial nerves examined.

Response: We have removed this comment from the manuscript and have now simply stated that tracer reaches the nasal cavity “Tracer was also apparent at the cribriform plate that separates the cranial and nasal cavities, indicating that outflow had also occurred along the olfactory nerves (cranial nerve I).” We hope that the new orientation figure (new Figure 3c) makes it clear that the perineural signals along the optical and trigeminal nerves were visualized at the base of the skull before they exit the cranium.

8. Citation in the Discussion (page 16, line 372) of reference 41, which was published in 1957, to document the presence of tight junctions in CNS blood vessels is invalid because tight junctions were not described as the basis of the barrier function of CNS blood vessels for another twelve years (J Cell Biol 40(3): 648-77, 1969).

Response: This citation was originally included as it clearly showed that tracers did not move from the interstitial tissue of the dura to the SAS. We agree with the reviewer that tight junctions had not yet been shown with electron microscopy to be the fundamental basis for CNS barriers until Brightman and Reese in 1969 and within the arachnoid layer until the work of Nabeshima et al, 1975.

Reviewer #2 (Remarks to the Author):

This is an ex vivo and in vivo study seeking to define lymphatic CSF outflow pathways from the cranium, and to define the effect of aging on the lymphatic clearance of CSF. The authors provide a detailed description of different perineural and other pathways by which CSF tracer exits the cranium, including providing some data that is at odds with findings from two recent papers^{1,2} describing tracer uptake into lymphatic vessels associated with the dural sinuses. Using both ex vivo and in vivo imaging of CSF tracer along lymphatic pathways, the authors provide evidence that lymphatic clearance of CSF is slowed in the aging versus the young adult brain. This is a well conceived and well written study, that makes an important contribution to the present understanding of lymphatic cerebrospinal fluid transport in the context of aging which may have important implications in the pathophysiology of neurodegenerative disease.

However, in several instances the conclusions drawn by the authors are somewhat overstated, or perhaps go beyond where the data that they provide would reasonably permit. These should be addressed, and are outlined in the comments below.

Response: We thank the reviewer for the kind comments.

Major Comments

1. In the results section, the authors state “The delay in the time for the signal to be apparent in the blood indicated that there did not appear to be direct venous uptake of the tracer, implying that arachnoid villi or other possible direct routes into the blood were not active under these conditions.” Throughout the study, the authors forcefully make the assertion that the slow appearance of CSF tracer into the saphenous blood pool indicates the absence of direct CSF-venous blood communication. An alternative interpretation of these data is that the delay of detectable tracer in the venous blood is due to dilution of the intravascular tracer below the authors’ detection limit. The authors’ conclusion regarding direct vascular uptake of the tracer should reflect this possibility, unless they can provide more direct evidence as to the lack of involvement of direct CSF-venous blood communication. Similarly, the editorial comments within the Results section related to these findings “...implying that arachnoid villi or other possible direct routes into the blood were not active...”, “In summary, the results of the dynamic imaging indicate that in mice the bulk of the outflow from the CSF of macromolecular and small molecular tracers was transported through the lymphatic system rather than venous routes”, etc. should be removed, and restricted to a more balanced treatment of this important issue in the Discussion.

Response: The reviewer is correct that we cannot absolutely rule out that there may be some direct blood flow within the first 25 min that is below the detection limit for our imaging setup. However, as shown in Supplemental Figure 4, the detection limit for P40D680 in the systemic blood is about 0.2% of the infused dose. If one assumes a steady rate of direct flow to blood for an entire hour there could not be more than 0.5% of the dose transported through this pathway, compared to the approximate 21.6% of the dose that reached the systemic blood at 60 min. To further support the claim of a lymphatic-predominant outflow of the tracer, we have now provided direct evidence of a lack of involvement of direct CSF-venous blood communication, namely through in vivo imaging of one the major collecting veins from the skull (new Supplemental Figure 5). These veins on each side of the mouse collect the blood from the transverse sinus that exits at the postglenoid foramen. At no point were we able to determine an increase in signal at this vein until it was obvious that a systemic blood increase had occurred. At this point (30 min after infusion), there was bright signal in the collecting lymphatics tracking towards both the mandibular and deep cervical lymph nodes, as well as obvious presence of signal at the saphenous vein (new Supplemental Figure 5). Also, in our investigations of the deep cervical region at 10 and 30 min there was never any apparent signal within the jugular veins. As mentioned above in our comments to Reviewer #1, we are unable to eliminate a scenario in which a direct blood outflow occurs at later time points. This possibility is now discussed. Throughout the manuscript we have been careful to state that blood outflow cannot be absolutely be ruled out and at no point do we make the claim that all outflow is lymphatic in nature.

2. The notion that lymphatic clearance of CSF is slowed in the aging brain is important, and has many critical implications for our understanding of age-related pathology, including neurodegenerative diseases; thus the observation that CSF tracer injected into the lateral ventricle appears more slowly in the lymphatic

drainage and saphenous blood pool is potentially very important. However, the movement of CSF tracer through the ventricular system to the blood pools is not dependent solely upon lymphatic clearance. It is also presumably influenced by the rate of CSF secretion, and bulk/dispersionary transport along the ventricular compartment. Given that CSF secretion is known to be reduced in the aging brain^{3,4}, it must be considered that the delayed appearance in the blood pool reflects the slowed transit through the ventricular system, versus the slowed clearance along the lymphatic vasculature. One way to address this would be to repeat these experiments using injection into a cisternal CSF compartment, thus bypassing the ventricular system.

Response: We agree with the reviewer that the movement of CSF tracer through the ventricular system to the systemic blood in aged mice may be a reflection of both slowed transit through the ventricles as well as diminished lymphatic clearance. Therefore, as the reviewer has suggested, we have repeated the aging study in mice after infusions into the cisterna magna (new Supplemental Figure 9). There was a decreased outflow of CSF in aged mice, albeit to lesser extent than after intraventricular infusion, indicating that slowed flow through the ventricles may in fact be occurring. However, a significant decline in total outflow with aging was still detected. As discussed above in our response to comment 8 of Reviewer 1, we now acknowledge in the Discussion other potential mechanisms for decreased CSF outflow in aged conditions.

3. In their discussion, the authors state “Our study is, to our knowledge, the first to characterize in detail the lymphatic outflow pathways of CSF from the skull in mice (Supplemental Fig. 7). Our use of pegylated NIR tracers and lymphatic-specific reporter mice combined with high-resolution stereomicroscopy have allowed an in-depth analysis that was not previously possible.” The use of a pegylated IR dye tracer in a lymphatic-specific reporter was previously employed by Aspleund et al., 2015 (DOI: 10.1084/jem.20142290) for exactly this purpose. Further, detailed pathways of lymphatic efflux from the central nervous system can be found in the literature throughout the last forty years, including many citations included within this manuscript. Therefore, the multiple statements throughout this manuscript, highlighting the novelty of these findings should be moderated.

Response: We thank the reviewer for the comment, however, we disagree that a systematic study outlining CSF outflow pathways in mice exists, as the reports by Aspelund et al. and Louveau et al. mostly focused on dural lymphatic vessels and to a lesser degree on the cribriform plate routes. Most previous studies have used larger animals and none have provided detailed images demonstrating the numerous perineural outflow locations and routing of CSF draining lymphatic vessels outside the skull. We have moderated the second sentence to “Our use of pegylated and small molecular NIR tracers and lymphatic-specific reporter mice combined with high-resolution stereomicroscopy have allowed an in-depth analysis of perineural outflow pathways and routing to CSF-draining lymph nodes.”

4. Within the introduction, the authors state, “Next, we used a recently developed tracer transport to blood assay to evaluate the dynamics of CSF outflow to the systemic circulation of a pegylated NIR dye macromolecular tracer in comparison to Evans blue, which behaves as a small molecule within the low protein CSF.” Evans blue avidly associates with albumin as discussed in the body of this manuscript, thus depending on the saturation of EB with albumin, a fraction of this tracer would behave more as a ~70 kDa protein, not a small molecule. Even if the presence of albumin in the CSF is lower than that of the blood, the 2.5 μ L injected may still be saturated. Do the authors know the fraction of the injected EB that is free vs. bound? If the authors wish to model the movement of a small molecule through the ventricular system and subarachnoid spaces, using an inert tracer such as a fluorescent dextran seemingly would be more appropriate.

Response: The reviewer is correct that we cannot rule out that some degree of albumin-binding of Evans blue occurs within the CSF and that a fraction of the tracer may behave like a ~70 kDa protein. Therefore, we have added data from infusions of two additional small molecular tracers, IRDye680 CW (which does not bind albumin but still exhibited binding to the arteries) and AlexaFluor 680-3kDa dextran (which does not bind albumin nor bind substantially to arteries, but exhibited some cellular uptake in the brain). We were unable to identify a small molecular tracer similar to P40D680 that does not show at least some retention in the brain. Regardless, as shown in new Figure 6, the data from all three small molecules indicated that there was a delayed uptake to systemic blood and signal within lymphatic vessels draining CSF.

Minor comments on figures

4d- Given that the authors collected dynamic imaging of bilateral deep cervical lymph nodes, it is unclear why the authors do not report mean enhancement +/- SD as in 4b.

4e- Please annotate this panel with the subset of lymph nodes under consideration.

Response: We have made it more clear that the noninvasive dynamic imaging of lymph nodes was performed on mandibular lymph nodes, rather than deep cervical lymph nodes. We have included a mean enhancement +/- SD in the mandibular lymph nodes in Figure 4d (new Figure 5d) and clearly indicated in the figure legend to new Figure 5e that the transit time to mandibular lymph nodes was compared to transit time to systemic blood.

1. Aspelund A, Antila S, Proulx ST, et al. A dural lymphatic vascular system that drains brain interstitial fluid and macromolecules. *J Exp Med.* 2015;212(7):991-999.
2. Louveau A, Smirnov I, Keyes TJ, et al. Structural and functional features of central nervous system lymphatic vessels. *Nature.* 2015;523(7560):337-341.
3. Stoquart-EISankari S, Baledent O, Gondry-Jouet C, Makki M, Godefroy O, Meyer ME. Aging effects on cerebral blood and cerebrospinal fluid flows. *J Cereb Blood Flow Metab.* 2007;27(9):1563-1572.

4. May C, Kaye JA, Atack JR, Schapiro MB, Friedland RP, Rapoport SI.
Cerebrospinal fluid production is reduced in healthy aging. Neurology. 1990;40(3 Pt 1):500-503.

Reviewers' comments:

Reviewer #1 (Remarks to the Author):

Review for Nature Communications of manuscript NCOMMS-17-00229 entitled "Outflow of cerebrospinal fluid is predominantly through lymphatic vessels in aged mice"

By Qiaoli Ma, Benjamin V. Ineichen, Michael Detmar and Steven T. Proulx

General comments:

1. The authors have effectively addressed most of the reviewer's comments and greatly improved the revised manuscript. Importantly, background information in the Introduction is more straightforward, and the Results are easier to understand.
2. There are, however, some problems that detract for the many attributes of the manuscript.
3. The authors claim in the Title, Abstract, and elsewhere in the manuscript that a major finding is that CSF outflow from the lateral ventricle is "predominantly" through lymphatic vessels. Although convincing evidence is provided for tracer movement from the lateral ventricle into lymphatics at multiple sites, evidence that lymphatics are the predominant route for CSF outflow is not compelling.
4. Outflow of CSF through venous blood:
 - a. The claim that little accumulation of tracer was detected in the posterior facial vein is not convincing evidence against significant CSF outflow through venous drainage from the brain.
 - b. The cited books from the 1960s on rat and mouse anatomy (references 39, 40) do not provide functional evidence of appreciable blood drainage from the brain through the posterior facial vein. Facial venous blood would be expected to dilute any blood from the brain.
 - c. If blood in the superior sagittal sinus cannot be directly measured, measurements of the tracer in blood of the internal jugular vein would be more compelling.
 - d. Without such evidence, the author's finding of slow tracer accumulation in the posterior facial vein (page 13, lines 296-301) is interpretable in relation to CSF outflow through venous blood.
5. Two approaches that could provide an acceptable solution:
 - a. Provide evidence that tracer measured in internal jugular venous blood comes exclusively from lymphatics and NOT from venous blood from the brain.
 - b. Eliminate all claims in the manuscript about the proportion of CSF drainage via lymphatics ("is predominantly through lymphatic vessels", "the major outflow pathway", "the major exit routes") in comparison to CSF drainage through other routes.
6. Absence of CSF drainage through meningeal lymphatics:
 - a. As the authors acknowledge, several groups have reported features of lymphatics in the meninges and have attributed functional significance to them.
 - b. Because of the important implications, the authors' observation that meningeal lymphatics did NOT contain detectable tracer after intraventricular injection should be supported by stronger evidence that can help to reconcile these observations with the other reports.
 - c. The authors' claim would be strengthened by better documentation that the meningeal lymphatics they observed were located in the dura, and that they were separated from the SAS by the arachnoidal barrier.
 - d. Unless the authors have evidence for another mechanism, the evidence presented should convincingly document that a diffusion barrier at the level of the arachnoid separates the SAS from

lymphatics in the dura.

e. Also, the comment about meningeal lymphatics forming a “discontinuous” network (Line 239) deserves better documentation, as it implies that meningeal lymphatics are unconventional.

Specific comments:

1. The authors description of the meningeal barriers is confusing or wrong (page 3, lines 71-75). It is unclear what is meant by “...leptomeninges, surrounding the SAS restrict flow of CSF into the brain parenchyma (pia mater...”. The SAS is bounded above by the arachnoid and below by the pia. Although the pia is not a permeability barrier between the SAS and the brain parenchyma, the arachnoid is a barrier between the SAS and the dura. The authors’ message here needs to be clarified. This would be helped by (a) consistent use of the terms dura, arachnoid, and pia; (b) specification of the location of the relevant barriers; and (c) avoiding the potentially confusing term “leptomeninges”.

2. A related source of confusion concerns the sheaths of cranial nerves (page 4, lines 99-102). The statement that “These sheaths are extensions of the SAS...” does not make sense. One is a connective tissue sheath, and the other is a space beneath a layer of the sheath. The space cannot be an extension of the connective tissue layer. The authors should clarify whether the concept is that the cranial nerve sheaths are extensions of the dura, arachnoid, and pia and that the SAS extends beneath the arachnoid-equivalent of cranial nerve sheaths.

3. The issue of the identity of the meningeal layers and barriers is also important in relation to the location of “meningeal lymphatics” (page 10, lines 229-234, and General Comment 6). Are these lymphatics located within the arachnoid, within the dura, or outside the dura (epidural)? How was this determined?

4. The implication that lymphatics could be relevant to the development of hydrocephalus needs clarification (page 5, lines 118-120). At the least, the authors should be clear about whether they are referring to internal hydrocephalus or external hydrocephalus. It is difficult to understand how impaired lymphatic drainage would contribute to internal hydrocephalus.

5. Page 10, Line 231: “...top of the skull”: “top” is not an anatomical term that unambiguously specifies a position or direction. What is the top of the spleen? Please use conventional anatomical positional and directional terms.

6. The statement about aging and dementia (page 15, lines 345-346) should be rewritten in more objective terms and moved to the Discussion. Perhaps that authors intended to say that neurodegenerative diseases, including Alzheimer’s disease and other dementias, increase with age.

7. Statements about reduced CSF outflow into blood with aging (page 15, line 362; page 17, lines 390-391) should be rewritten in more objective terms in the context of the data. Was the outflow volume reduced? Was the rate of CSF tracer accumulation in blood slower in older mice? If it was the latter, it should be described as such, to avoid the ambiguity of “reduced CSF outflow”.

8. Please replace informal news report-like generalizations with objective language that refers to the literature. Two examples are: (1) “has traditionally been considered”, Line 67, would be better as “The choroid plexus is a major source of cerebrospinal fluid (CSF) that circulates through the ventricular system and subarachnoid space (authoritative references).” (2) “is accepted to take place”, Line 82, would be better as “Arachnoid villi have been considered an important site of outflow of the CSF into venous blood (authoritative references).”

9. Numbered figures would help the reviewer.

Reviewer #2 (Remarks to the Author):

The authors have substantially revised their initial submission, addressing each concern that was raised in the initial review. This has included both a muting of some of the less concretely founded conclusions in the initial submission, as well as some of the novelty claims. The addition of the intracisternal infusion series has added an important new detail to the study, and has confirmed that the effect of aging on CSF clearance is likely not due simply to changes in the rate of CSF secretion with age.

With these changes, and with the changes made in response to the other referee, I have no further critiques.

Response to reviewer comments:

Reviewer #1 (Remarks to the Author):

General comments:

1. The authors have effectively addressed most of the reviewer's comments and greatly improved the revised manuscript. Importantly, background information in the Introduction is more straightforward, and the Results are easier to understand.

Response: We thank the reviewer for the kind comments.

2. There are, however, some problems that detract for the many attributes of the manuscript.

3. The authors claim in the Title, Abstract, and elsewhere in the manuscript that a major finding is that CSF outflow from the lateral ventricle is "predominantly" through lymphatic vessels. Although convincing evidence is provided for tracer movement from the lateral ventricle into lymphatics at multiple sites, evidence that lymphatics are the predominant route for CSF outflow is not compelling.

4. Outflow of CSF through venous blood:

a. The claim that little accumulation of tracer was detected in the posterior facial vein is not convincing evidence against significant CSF outflow through venous drainage from the brain.

b. The cited books from the 1960s on rat and mouse anatomy (references 39, 40) do not provide functional evidence of appreciable blood drainage from the brain through the posterior facial vein. Facial venous blood would be expected to dilute any blood from the brain.

c. If blood in the superior sagittal sinus cannot be directly measured, measurements of the tracer in blood of the internal jugular vein would be more compelling.

d. Without such evidence, the author's finding of slow tracer accumulation in the posterior facial vein (page 13, lines 296-301) is interpretable in relation to CSF outflow through venous blood.

5. Two approaches that could provide an acceptable solution:

a. Provide evidence that tracer measured in internal jugular venous blood comes exclusively from lymphatics and NOT from venous blood from the brain.

b. Eliminate all claims in the manuscript about the proportion of CSF drainage via lymphatics ("is predominantly through lymphatic vessels", "the major outflow pathway", "the major exit routes") in comparison to CSF drainage through other routes.

Response: We disagree with the reviewer's assessment that the posterior facial vein is not an appropriate location to measure direct venous outflow from the skull in the mouse.

While the internal jugular vein would be the most suitable location for these measurements in humans and in large animals, in the mouse this vein was shown in a thorough imaging study to be much smaller than the external jugular vein and to receive only a minor portion of the total cerebral blood volume (see Mancini et al., PloS One, 2015, Figs. 8, 14, 16 and 19). Instead, the blood from the superior sagittal and transverse sinuses, as well as the majority of the blood from the brain, drains into vessels that connect to the posterior facial vein and eventually the external jugular vein.

Since some venous outflow, particularly from the caudal region, does occur through the internal jugular vein, we nonetheless explored the potential to perform measurements in this region. However, due to its close proximity to the cervical lymphatics it is impossible to measure with the sensitivity required to detect signal in the internal jugular vein. Using the same microscope and image settings used to image the venous signals at either the posterior facial vein or saphenous vein regions, the images are oversaturated by signal from the tracer-filled lymphatic vessels and deep cervical nodes and this signal bleeds into the region over the internal jugular vein. These images also nicely demonstrate the extreme differences in tracer concentration within the lymphatic system in comparison to venous blood.

In response to comment 4d, it is highly unlikely that the signal that is being detected at the later time points in the posterior facial vein is evidence of a direct blood outflow. The time point of the initial increase as well as the overall increase in signal at 30 min, are directly comparable to the values detected at the saphenous vein that are shown in either Figures 5b and 7c (young group). Imaging of these two veins was acquired at almost identical image settings with only a slight difference in the zoom factor. Even if tracer-containing blood coming from the transverse sinuses were diluted by the blood coming from facial vein, the signal within the posterior facial vein should still be far higher than tracer signal that is diluted throughout the entire body.

We have already addressed this limitation of the study in the discussion by acknowledging that we cannot completely rule out that some direct blood uptake could occur at later time points. However, we strongly feel that with the evidence that we have presented, as well as the discussion of previous studies on 5 different species that could not detect direct venous outflow in either the superior sagittal sinus or internal jugular

vein (with some exceptions at high intracranial pressures), we are fully justified in concluding that the CSF is predominantly drained by the lymphatic system.

Additional reference: Mancini M, Greco A, Tedeschi E, et al. Head and Neck Veins of the Mouse. A Magnetic Resonance, Micro Computed Tomography and High Frequency Color Doppler Ultrasound Study. *Sen U*, ed. *PLoS ONE*. 2015;10(6):e0129912. doi:10.1371/journal.pone.0129912.

6. Absence of CSF drainage through meningeal lymphatics:

- a. As the authors acknowledge, several groups have reported features of lymphatics in the meninges and have attributed functional significance to them.*
- b. Because of the important implications, the authors' observation that meningeal lymphatics did NOT contain detectable tracer after intraventricular injection should be supported by stronger evidence that can help to reconcile these observations with the other reports.*

Response: The reviewer is correct in the assertion that our results differ from the previous reports that were able to show uptake into dural lymphatic vessels. As far as we can determine, four previous studies have demonstrated at least some uptake of CSF- or brain-injected tracers into lymphatic vessels located in the dura mater. The first of these studies, by Butler et al., 1984 (Ref. 49), found horseradish peroxidase (HRP) in vessels that they identify as “initial lymphatics” in the dura mater of the cat only under high intracranial pressures. They attribute this to a opening of the arachnoid layer that was clearly a barrier for transport of HRP from the CSF to the extracellular tissue of the dura at lower pressures. The second study, Kida et al., 1993 (Ref. 21), showed evidence of India ink particles in dural lymphatic vessels in rat, which they deemed of minor importance in comparison to an exit route through the cribriform plate. In this study it is important to note that they injected 50 μ L into the cisterna magna, which is approximately one-fifth of the total CSF volume in the rat.

The most recent reports by Aspelund et al., 2015 (Ref. 31) and Louveau et al., 2015 (Ref. 30) are more bold in their assertions that the dural lymphatic vessels are important outflow pathways, however, neither report acknowledges the presence of the arachnoid barrier. The Aspelund et al. report demonstrated the presence of a 20 kDa pegylated tracer injected into the brain parenchyma within the dura mater lymphatics only at the base of the skull. There was also increased filling of the network after ligation of deep cervical lymph nodes. This implies that at least some of this filling may have been from retrogradal flow, perhaps from the perineural outflow route at the jugular foramen that is demonstrated in the current study. Interestingly, retrograde flow of lymph into dural lymphatic vessels after ligation of cervical lymphatics in dogs precisely at the jugular foramen location was shown in an early report by Földi et al., 1966 (Ref. 48). There is also the possibility that alternate routes exist for flow from the brain ISF to reach the dura mater rather than from CSF.

This leaves the report by Louveau et al., 2015 which showed evidence of uptake of quantum dots and Evans blue within the lymphatic vessels. We also used Evans blue in

the current study but were unable to detect any uptake within the dural lymphatic vessels (data not shown). This could be due to the lower dose that we used (2.5 μ L of 0.6% compared to 5 μ L of 10% Evans blue), however, we have seen obvious neurotoxic effects (seizures) when concentrations above 2.5% were injected into the lateral ventricle. We are unable to explain the discrepancy that exists between the quantum dot data from Louveau et al., 2015 and our data using pegylated dyes. We do not use quantum dots in our laboratory as they are readily phagocytosed and are retained in several locations within the mouse, including lymph nodes. Therefore, in our opinion, they are not ideal tracers for lymphatic function assessments, but as a macromolecular tracer they should have been prevented from accessing the dural lymphatic vessels unless the injection protocol induced a disruption of the arachnoid barrier. Several details are missing from the Louveau et al. report including what rate the injections were performed into the cisterna magna, what time points after injection the imaging was performed or what concentration of quantum dots was injected.

Since we fully acknowledge that our data cannot completely exclude that there may be uptake by dural lymphatic vessels, we have left the relative contribution of these vessels towards the total CSF outflow as an open question. A recently published review by Coles et al, *Progress in Neurobiology*, 2017 has also questioned this pathway from CSF to the dural lymphatics: “How the dural lymph vessels participate in the CSF circulation is unknown: Louveau et al. (2015) simply shown an arrow crossing the arachnoid membrane, which is thought to be impermeable (their ED Fig.10).”

Additional reference:

Coles, J.A., Myburgh, E., Brewer, J.M., McMenamin, P.G., Where are we? The anatomy of the murine cortical meninges revisited for intravital imaging, immunology, and clearance of waste from the brain., *Progress in Neurobiology* (2017), <http://dx.doi.org/10.1016/j.pneurobio.2017.05.002>

c. The authors' claim would be strengthened by better documentation that the meningeal lymphatics they observed were located in the dura, and that they were separated from the SAS by the arachnoidal barrier.

d. Unless the authors have evidence for another mechanism, the evidence presented should convincingly document that a diffusion barrier at the level of the arachnoid separates the SAS from lymphatics in the dura.

Response: There is widespread agreement in the field that the meningeal lymphatic vessels are located within the dura mater. The only dissenting viewpoint appears to be that of the Kipnis group, who without any anatomical evidence give two other potential locations of these vessels in an opinion article published last year (Raper et al, *Trends in Neuroscience*, 2016). The dura mater is extensively vascularized with fenestrated blood vessels, so it is only logical that it would contain a lymphatic network. On the other hand, the arachnoid is avascular. The arachnoid barrier layer has already been demonstrated in mice in the original report from Nabeshima et al, 1975 (Ref. 4). For an extensive presentation of the evidence for the localization of the lymphatic vessels in the dura mater and the presence of a barrier layer at the arachnoid, we again refer the reviewer to the just

published review by Coles et al, *Progress in Neurobiology*, 2017. To avoid any potential confusion regarding this issue, we have avoided all uses of the term “meningeal” lymphatics and have exclusively used the term “dural” lymphatic vessels.

Additional reference: Raper, D., Louveau, A., & Kipnis, J. (2016). How Do Meningeal Lymphatic Vessels Drain the CNS? *Trends in Neurosciences*, 39(9), 581–586.
<http://doi.org/10.1016/j.tins.2016.07.001>

e. Also, the comment about meningeal lymphatics forming a “discontinuous” network (Line 239) deserves better documentation, as it implies that meningeal lymphatics are unconventional.

Response: We have added additional images to Supplemental Figure 3 to demonstrate the discontinuous nature of the dural lymphatic network near the superior sagittal sinus. It is already known that this network is unconventional since the lymphatic vessels lack intraluminal valves and it is not organized into a plexus typical of initial lymphatic networks in other organs.

Specific comments:

1. The authors description of the meningeal barriers is confusing or wrong (page 3, lines 71-75). It is unclear what is meant by “...leptomeninges, surrounding the SAS restrict flow of CSF into the brain parenchyma (pia mater...”. The SAS is bounded above by the arachnoid and below by the pia. Although the pia is not a permeability barrier between the SAS and the brain parenchyma, the arachnoid is a barrier between the SAS and the dura. The authors’ message here needs to be clarified. This would be helped by (a) consistent use of the terms dura, arachnoid, and pia; (b) specification of the location of the relevant barriers; and (c) avoiding the potentially confusing term “leptomeninges”.

Response: We agree that the description of the meningeal barriers may have been confusing but it was not wrong. Leptomeninges is a commonly used term in the field to describe the arachnoid and pia mater. Regardless, we have removed the mention of this term and restructured this section of the introduction to first describe the location of the meningeal layers and, subsequently, the barrier functions.

2. A related source of confusion concerns the sheaths of cranial nerves (page 4, lines 99-102). The statement that “These sheaths are extensions of the SAS...” does not make sense. One is a connective tissue sheath, and the other is a space beneath a layer of the sheath. The space cannot be an extension of the connective tissue layer. The authors should clarify whether the concept is that the cranial nerve sheaths are extensions of the dura, arachnoid, and pia and that the SAS extends beneath the arachnoid-equivalent of cranial nerve sheaths.

Response: The reviewer is correct in pointing out that this was not written correctly. We intended to state that “These sheaths enclose extensions of the SAS...”. This has been

changed in the text.

3. The issue of the identity of the meningeal layers and barriers is also important in relation to the location of “meningeal lymphatics” (page 10, lines 229-234, and General Comment 6). Are these lymphatics located within the arachnoid, within the dura, or outside the dura (epidural)? How was this determined?

Response: As discussed above, the lymphatic vessels of the meningeal layers lining the skull have been clearly identified in several previous studies to be located within the dura mater. This is the only possible location as the arachnoid membrane is avascular and there is no “epidural” space within the skull cavity. To avoid any potential confusion, we have avoided all uses of the term “meningeal” lymphatics and have exclusively used the term “dural” lymphatic vessels.

4. The implication that lymphatics could be relevant to the development of hydrocephalus needs clarification (page 5, lines 118-120). At the least, the authors should be clear about whether they are referring to internal hydrocephalus or external hydrocephalus. It is difficult to understand how impaired lymphatic drainage would contribute to internal hydrocephalus.

Response: We realize that at this point implicating the lymphatic system to the development of hydrocephalus is speculation. Therefore, we have removed this statement from the manuscript.

5. Page 10, Line 231: “...top of the skull”: “top” is not an anatomical term that unambiguously specifies a position or direction. What is the top of the spleen? Please use conventional anatomical positional and directional terms.

Response: We thank the reviewer for pointing out the inadvertent use of a non-anatomical term. We have now changed this to “coronal aspect of the skull.”

6. The statement about aging and dementia (page 15, lines 345-346) should be rewritten in more objective terms and moved to the Discussion. Perhaps that authors intended to say that neurodegenerative diseases, including Alzheimer’s disease and other dementias, increase with age.

Response: It is well-known that the prevalence of Alzheimer’s disease and dementia increase with advancing age. This statement is necessary at this point of the results to introduce the justification for undertaking the aging study.

7. Statements about reduced CSF outflow into blood with aging (page 15, line 362; page 17, lines 390-391) should be rewritten in more objective terms in the context of the data. Was the outflow volume reduced? Was the rate of CSF tracer accumulation in blood slower in older mice? If it was the latter, it should be described as such, to avoid the ambiguity of “reduced CSF outflow”.

Response: We have clarified these statements to state that “the dynamics of CSF outflow to blood were slower in aged mice” and “a significant delay in lymphatic outflow from CSF in aged mice.”

8. Please replace informal news report-like generalizations with objective language that refers to the literature. Two examples are: (1) “has traditionally been considered”, Line 67, would be better as “The choroid plexus is a major source of cerebrospinal fluid (CSF) that circulates through the ventricular system and subarachnoid space (authoritative references).” (2) “is accepted to take place”, Line 82, would be better as “Arachnoid villi have been considered an important site of outflow of the CSF into venous blood (authoritative references).”

Response: As this paper is focused on the outflow of CSF, we do not feel it would be appropriate to begin the introduction to the paper with “The choroid plexus is...”. We have instead changed “has traditionally been considered” to “is considered”.

9. Numbered figures would help the reviewer.

Response: For the resubmission, it was not possible to incorporate the figures into one PDF for the reviewer with labeled figures and legends as the journal requests high quality versions of the figures that is assembled into a PDF automatically.

Reviewer #2 (Remarks to the Author):

The authors have substantially revised their initial submission, addressing each concern that was raised in the initial review. This has included both a muting of some of the less concretely founded conclusions in the initial submission, as well as some of the novelty claims. The addition of the intracisternal infusion series has added an important new detail to the study, and has confirmed that the effect of aging on CSF clearance is likely not due simply to changes in the rate of CSF secretion with age.

With these changes, and with the changes made in response to the other referee, I have no further critiques.

Response: We thank the reviewer for the kind comments and fair review.

REVIEWERS' COMMENTS:

Reviewer #1 (Remarks to the Author):

Review of Revision #2 for Nature Communications of manuscript NCOMMS-17-00229 entitled "Outflow of cerebrospinal fluid is predominantly through lymphatic vessels and is reduced in aged mice"

By Qiaoli Ma, Benjamin V. Ineichen, Michael Detmar and Steven T. Proulx

General Comments:

1. In this second revision, the authors have made numerous additional improvements to their manuscript and have adequately addressed many of the issues raised about Revision #1.
2. The rebuttals set out in the lengthy point-by-point response accompanying Revision #2 explain the authors' rationale for making or not making the changes recommended in the previous review. Most of these are fine, but a few substantive issues deserve more attention.
3. The authors present solid evidence for appreciable CSF outflow into the bloodstream through lymphatics. This evidence fits with and nicely complements many previous reports in the literature.
4. However, the claims in the title that CSF outflow occurs "predominately" through lymphatics and in the abstract and main body of the manuscript (pages 2, 18, 21, 23, 24) that lymphatics are the major outflow pathway are not convincingly supported by the evidence presented in the manuscript.
5. The authors' data document that lymphatics are a major pathway but not the major pathway.
6. Measurements of the posterior facial vein are relevant because they reflect one potential CSF outflow route, but they do not exclude the contributions other CSF drainage routes that have been reported. Absence of evidence is not evidence of absence. The authors did not address the contributions of the tributaries of the internal jugular vein, diploic veins, epidural veins, spinal veins, choroid plexus, and other reported routes of CSF outflow.
7. To determine whether lymphatics are the predominate route for CSF outflow, a mass balance analysis would be needed to weigh quantitatively the outflow through lymphatics in proportion to total CSF outflow, as reported by others (e.g., Boulton et al. AmJPhysiol 1998, 1999).
8. A simple solution is to delete "predominantly" from the title and describe lymphatics throughout the text as "a" major pathway instead of "the" major pathway of CSF outflow.
9. As Miles Johnston and colleagues wrote in one of their 30-plus papers on CSF outflow through lymphatics, which include analyses of CSF mass balance, "While the definitive judgment on the proportional CSF clearance that occurs through the various potential pathways has yet to be written, a pattern seems to be emerging: one that assigns a major role to extracranial lymphatic vessels..." (Zakharov et al. Microvascular Res 2004). This is a prudent approach for the authors to emulate, as it leaves room for future studies by them and others to further the understanding of CSF clearance.
10. A harmonious example of an alternative title that acknowledges the importance of both lymphatics and aging and is consistent with the authors' data is: "Major contribution of lymphatics to cerebrospinal fluid outflow is reduced in aged mice"

Specific Comments:

1. The paper by Mancini et al. 2015, cited as reference 41, is relevant to the interpretation of the authors' fluorescence measurements of the posterior facial vein, but in fairness, the authors should inform readers of the limitations of this study using Microfil to characterize the vascular anatomy and other caveats acknowledged by Mancini et al. The authors' statement (pages 12-13), "In rodents, the major venous outflow route for blood from the brain and the dural sinuses ...drains into the posterior facial vein to reach the external jugular vein, as opposed to the internal jugular vein..." should be described in more objective terms in the context of the report by Mancini et al., e.g., "Although the venous drainage of the dural sinuses has not been specifically assessed in mice, Microfil casting of the cerebral vasculature revealed that the olfactory bulbs and frontal, parietal, and temporal lobes of the brain drain mainly through the external jugular veins, whereas the occipital lobe and cerebellum drain mainly through the internal jugular (reference 41)."
2. The authors should cite additional experimental evidence, if available, to support their view of the venous drainage of dural sinuses in mice.
3. Throughout manuscript: "paravascular" should be replaced with "perivascular".
4. The authors' references to the existence of consensus and how others view issues is presumptuous. Please change the following to reflect the strength of the evidence or specify how the authors interpret the evidence rather than the authors' assumption of how others view the evidence:
 - a. Abstract: "commonly accepted"
 - b. Page 3: "is considered to be"
 - c. Page 3: "has not led to a consensus"
 - d. Page 3: "are believed to be absent"
 - e. Page 3: "were initially believed"
 - f. Page 4: "widespread acceptance"
 - g. Page 5: "there is a lack of consensus on"
 - h. Page 7: "CSF is considered to be"
 - i. Page 11: "then one would expect"
5. Abstract, page 2: "...suggesting that the lymphatic system may represent a target for age-associated neurological conditions." The implication of this statement in the context of targeting lymphatics in neurodegenerative diseases is unclear. Please clarify and explain the concept in the Discussion (see Specific Comment #35).
6. Introduction, page 3: "The CSF in the SAS is contained within the pia mater, which is semi-permeable, and the arachnoid..." Do the authors mean, "CSF flows through the subarachnoid space (SAS) between the arachnoid and the pia..."?
7. Introduction, page 3, "interstitial tissue": It is unclear what the authors mean by interstitial tissue in the brain. Perhaps the authors mean "interstitial space" or "interstitial fluid".
8. Introduction, page 4, "some species such as rabbit and sheep": The rat should be added to these species (Boulton et al. AmJPhysiol 1999).
9. Introduction, page 4, "extensions of the SAS that project extracranially": Has the route between the SAS and extracranial nerves been identified? Describe the evidence that the SAS projects extracranially.

10. Introduction, page 5, "the current paradigm suggests dual-outflow pathways for CSF": It would be more accurate to describe the current status as "two or more outflow pathways".

11. Results, page 7 and elsewhere, "sacrificed": "euthanized" would be better.

12. Results, page 7, "tracer appeared to localize along the paravascular spaces of arteries": Because the exact location of the tracer was not determined, it would be better to say, "tracer accumulated around arteries" or "tracer stained the adventitial surface of arteries"

13. Results, page 7, "'arteries, such as the middle cerebral artery, over the convexities of the cortical hemispheres": It would be better to refer to these vessels as "branches of the middle cerebral artery...".

14. Results, page 8, "There was also entry of tracer into paravascular spaces around penetrating arteries": I recommend, "Tracer was also found in the Virchow-Robin space around penetrating arteries". As it surrounds the vessels, this is a "perivascular" space not a "paravascular" space.

15. Results, page 9, "surgically exposed": Because this implies that the mouse was alive during surgery, it would be better to delete "surgically".

16. Results, page 12, "The delay in the time for the signal to be apparent in the blood suggested that there did not appear to be rapid venous uptake of the tracer, implying that direct routes into the blood may not be active under these conditions.": This statement makes the assumption that a direct route would be faster than via lymphatics, but no rationale or justification is given. A more accurate statement would be that a 25-minute delay was found between tracer infusion into the CSF and detection in the saphenous vein regardless of the route.

17. Results, page 13, "In sum, the dynamic imaging approaches indicate that active lymph transport outside the skull rather than direct blood outflow was primarily responsible": This statement should be revised along the following lines: "Based on our dynamic imaging data, we conclude that lymph transport was the main route for tracer movement from the CSF into the bloodstream, and that outflow through the posterior facial vein had little contribution."

18. Results, page 14, "Surprisingly, we could not detect immediate blood uptake of any tracer, with the patterns exhibiting delays before signal could be detected peripherally": Why is this surprising? These tracers do not leak from the blood into most regions of the brain, so why would movement from the brain into blood be expected? Please modify accordingly.

19. Results, page 15, "the CSF outflow routes and the dynamics of transport to systemic blood were similar": This statement should be revised along the following lines: "the CSF outflow routes through lymphatics and the dynamics of transport to systemic blood were similar"

20. Results, page 15, "Aging is closely associated with the development of several neurological disorders such as Alzheimer's disease and dementia." The link between neurodegenerative diseases and aging would be more clear if described as, "The incidence of Alzheimer's disease and other dementias and many other neurodegenerative diseases increases with age."

21. Results, page 15, "hypothesis for the development of these disorders has been developed that proposes": More straightforward: "hypothesis for the development of these disorders was proposed"

22. Results, page 16, "However, no significant difference could be seen at 60 min indicating a delayed

outflow pattern in aged mice." The reasoning here is unclear. If the slope was less steep in aged mice, shouldn't the accumulation in these mice be less at 60 minutes? Please modify accordingly.

23. Results, page 17, "...indicated a significant delay in lymphatic outflow from CSF in aged mice." The word "delay" is confusing. The data seem to fit better with, "...indicated significantly slower CSF outflow into lymphatics of aged mice." Please modify accordingly.

24. Discussion, page 18, "Our study is, to our knowledge, the first...": This would be a good place to cite the recent paper on fluorescence imaging of lymphatic outflow of CSF in mice by Kwon et al. (J Immunol Methods, 2017).

25. Discussion, pages 19-20, "We found that these lymphatic vessels are frequently discontinuous, have small diameters and, as previously reported, lack intraluminal valves (except at the base of the skull), indicating that they are different from conventional lymphatic networks." Data from the present study should be referred to in the past tense in the Discussion.

26. Discussion, page 20, "Since the blood vessels of the dura mater tissue lack tight junctions (reference 4), it is possible that dural lymphatic vessels exist to drain this tissue specifically.": This statement is incorrect. Reference 4 reports that some dural blood vessels have endothelial fenestrations, but does not describe the absence of tight junctions in these endothelial cells. It is unlikely that endothelial cells of any blood vessels lack tight junctions. Please modify accordingly.

27. Discussion, page 20, "dural lymphatic vessels in perineurial tissue of the optic and facial nerves..": Does this mean, "lymphatic vessels in the dura around the optic and facial nerves"? As the perineurium is one of the three sheaths around peripheral nerves, "perineurial tissue" could be misinterpreted.

28. Discussion, page 20, "Clearly, more research is needed to determine the importance of the dural lymphatic route for CSF outflow in comparison to the more established perineural pathways to reach extracranial lymphatic vessels." This sentence would better fit the authors' findings if revised as follows: "Clearly, the function of dural lymphatics deserves more research to determine whether they contribute to CSF outflow to the extent found for lymphatic vessels associated with cranial nerves and spinal nerves."

29. Discussion, page 21, "suggest that lymphatic transport is the predominant CSF outflow pathway": "the predominant" should be changed to "an important" or "a major". See General Comments 4-8.

30. Discussion, page 21, "a lymphatic-predominant drainage of CSF": Please revise as described in previous comment.

31. Discussion, page 22, "since a continuous lining of endothelial cells with tight junctions was found to exist on the villus": The authors' argument against CSF movement into dural sinuses based on the presence of tight junctions should be tempered because the same argument applies to CSF movement from the SAS into lymphatics where tight junctions in the arachnoid barrier separate the two.

32. Discussion, page 23, "To our knowledge, our study is the first to demonstrate a reduction of lymphatic outflow of CSF with aging.": This statement should be modified to accommodate a similar statement made 10 years ago in the paper cited as reference 69, which initially described the reduction in CSF outflow with aging.

33. Discussion, page 23, "An additional explanation for the reduced CSF outflow": This would be a

good place to acknowledge the age-related functional changes in the superior sagittal sinus that have been found in mice by Kang et al. (Neurobiol Aging, 2016).

34. Conclusions, page 24, "the major exit routes": See General comments 4-8.

35. Conclusions, page 24, "there may be potential for clinical translation of the imaging technique, which could allow noninvasive monitoring of CSF outflow in patients suffering from neurological disorders": As for the final sentence of the Abstract, the final sentence of the conclusions needs additional explanation in the Discussion to enable readers to understand what the authors are suggesting.

36. Acknowledgements, page 35: "sharing of animal licenses". The meaning of this statement could be misinterpreted in a negative way, where readers could think that the authors circumvented the animal licensing policy. Please clarify.

Response to reviewer comments:

Reviewer #1 (Remarks to the Author):

Review of Revision #2 for Nature Communications of manuscript NCOMMS-17-00229 entitled "Outflow of cerebrospinal fluid is predominantly through lymphatic vessels and is reduced in aged mice"

By Qiaoli Ma, Benjamin V. Ineichen, Michael Detmar and Steven T. Proulx

General Comments:

- 1. In this second revision, the authors have made numerous additional improvements to their manuscript and have adequately addressed many of the issues raised about Revision #1.*
- 2. The rebuttals set out in the lengthy point-by-point response accompanying Revision #2 explain the authors' rationale for making or not making the changes recommended in the previous review. Most of these are fine, but a few substantive issues deserve more attention.*
- 3. The authors present solid evidence for appreciable CSF outflow into the bloodstream through lymphatics. This evidence fits with and nicely complements many previous reports in the literature.*

Response: We thank the reviewer for the kind comments.

- 4. However, the claims in the title that CSF outflow occurs "predominately" through lymphatics and in the abstract and main body of the manuscript (pages 2, 18, 21, 23, 24) that lymphatics are the major outflow pathway are not convincingly supported by the evidence presented in the manuscript.*
- 5. The authors' data document that lymphatics are a major pathway but not the major pathway.*
- 6. Measurements of the posterior facial vein are relevant because they reflect one potential CSF outflow route, but they do not exclude the contributions other CSF drainage routes that have been reported. Absence of evidence is not evidence of absence. The authors did not address the contributions of the tributaries of the internal jugular vein, diploic veins, epidural veins, spinal veins, choroid plexus, and other reported routes of CSF outflow.*
- 7. To determine whether lymphatics are the predominate route for CSF outflow, a mass balance analysis would be needed to weigh quantitatively the outflow through lymphatics in proportion to total CSF outflow, as reported by others (e.g., Boulton et al. AmJPhysiol 1998, 1999).*

8. A simple solution is to delete “predominantly” from the title and describe lymphatics throughout the text as “a” major pathway instead of “the” major pathway of CSF outflow.

9. As Miles Johnston and colleagues wrote in one of their 30-plus papers on CSF outflow through lymphatics, which include analyses of CSF mass balance, “While the definitive judgment on the proportional CSF clearance that occurs through the various potential pathways has yet to be written, a pattern seems to be emerging: one that assigns a major role to extracranial lymphatic vessels...” (Zakharov et al. *Microvascular Res* 2004). This is a prudent approach for the authors to emulate, as it leaves room for future studies by them and others to further the understanding of CSF clearance.

10. A harmonious example of an alternative title that acknowledges the importance of both lymphatics and aging and is consistent with the authors’ data is: “Major contribution of lymphatics to cerebrospinal fluid outflow is reduced in aged mice”

Response: It appears that we will need to agree to disagree on this point. We were asked to demonstrate direct imaging of a collecting vein in the first round of review. We provided this data for the posterior facial vein and were unable to show any direct venous uptake of our tracer (despite the assay being sensitive enough to detect low levels of tracer diluted in systemic blood). As discussed in the second round of revisions and again below, the posterior facial vein is the most appropriate location in rodents to test for venous uptake from the dural sinuses. Testing for direct venous uptake at the multiple other locations suggested by the reviewer would be an exercise in futility. Likewise, a mass balance approach would be impossible in mice as it requires all the collecting lymphatic vessels draining CSF to be cannulated, which was not possible even in larger species such as sheep, as fully acknowledged within the reports by the group of Miles Johnston. Our novel dynamic imaging approach shows clearly that the major (if not sole) contributor of the tracer transport to the systemic blood is the lymphatic system and not collecting veins.

Specific Comments:

1. The paper by Mancini et al. 2015, cited as reference 41, is relevant to the interpretation of the authors’ fluorescence measurements of the posterior facial vein, but in fairness, the authors should inform readers of the limitations of this study using *Microfil* to characterize the vascular anatomy and other caveats acknowledged by Mancini et al. The authors’ statement (pages 12-13), “In rodents, the major venous outflow route for blood from the brain and the dural sinuses ...drains into the posterior facial vein to reach the external jugular vein, as opposed to the internal jugular vein...” should be described in more objective terms in the context of the report by Mancini et al., e.g., “Although the venous drainage of the dural sinuses has not been specifically assessed in mice, *Microfil*

casting of the cerebral vasculature revealed that the olfactory bulbs and frontal, parietal, and temporal lobes of the brain drain mainly through the external jugular veins, whereas the occipital lobe and cerebellum drain mainly through the internal jugular (reference 41).”

2. The authors should cite additional experimental evidence, if available, to support their view of the venous drainage of dural sinuses in mice.

Response: The Mancini et al report uses additional imaging techniques beyond Microfil casting including magnetic resonance angiography and Doppler ultrasound to demonstrate the venous outflow pathways from the murine cranium. Detailed descriptions of the drainage routes of the dural sinuses (superior sagittal sinus, transverse sinus, straight sinus, sigmoid sinus, etc.) are also included in this report. The paper concludes that the transverse to petrosquamous sinus route draining to the posterior facial vein and external jugular vein is the major route rather than outflow to the internal jugular vein. This is consistent with the older anatomical works on mouse and rat that are also cited (Refs. 39 and 40). Therefore, the statement as written in the manuscript is valid.

3. Throughout manuscript: “paravascular” should be replaced with “perivascular”.

Response: The terminology in the literature (“perivascular” vs. “paravascular”) for flow around arteries and veins in the CNS is inconsistent. In accordance with Engelhardt et al., Acta Neuropathol, 2016 who attempted to clarify the difference between the two terms, we purposely used “paravascular” rather than “perivascular” to denote the presence of the 40 kDa tracer at the outer aspects of the arteries and veins lining the subarachnoid space.

4. The authors’ references to the existence of consensus and how others view issues is presumptuous. Please change the following to reflect the strength of the evidence or specify how the authors interpret the evidence rather than the authors’ assumption of how others view the evidence:

- a. Abstract: “commonly accepted”*
- b. Page 3: “is considered to be”*
- c. Page 3: “has not led to a consensus”*
- d. Page 3: “are believed to be absent”*
- e. Page 3: “were initially believed”*
- f. Page 4: “widespread acceptance”*
- g. Page 5: “there is a lack of consensus on”*
- h. Page 7: “CSF is considered to be”*
- i. Page 11: “then one would expect”*

Response: We disagree that these types of statements are presumptuous. For example, it is more appropriate to write “Within the CNS itself, lymphatic vessels are believed to be absent” rather than “Within the CNS itself, lymphatic vessels are absent” as this leaves room for future discoveries that may change this paradigm. We do agree that writing “then one would expect” is not ideal, so we have deleted this statement on Page 11.

5. *Abstract, page 2: "...suggesting that the lymphatic system may represent a target for age-associated neurological conditions." The implication of this statement in the context of targeting lymphatics in neurodegenerative diseases is unclear. Please clarify and explain the concept in the Discussion (see Specific Comment #35).*

Response: We have already clarified this in the Results and Discussion. From the Results section explaining the justification for the aging studies: "The incidence of Alzheimer's disease and other dementias and many other neurodegenerative diseases increases with age. Recently, a hypothesis for the development of these disorders was proposed that toxic proteins such as amyloid beta and tau may accumulate in the brain due to reduced clearance (Ref. 42). Earlier studies have shown a reduced turnover of CSF and removal of labeled proteins, including amyloid beta, after ventricular-cisternal perfusion in aged rats (Ref. 43)." From the Discussion: "It will be interesting to test whether a more accelerated decrease in CSF lymphatic outflow exists in mouse models of Alzheimer's disease and whether a functional decline is associated with the development of amyloid beta plaques. If so, the lymphatic system may represent a possible new therapeutic target with the aim to enhance the clearance of toxic proteins from the CSF and the brain."

6. *Introduction, page 3: "The CSF in the SAS is contained within the pia mater, which is semi-permeable, and the arachnoid..." Do the authors mean, "CSF flows through the subarachnoid space (SAS) between the arachnoid and the pia..."?*

Response: We agree that "flows through" would be better choice than "is contained within". This has been modified in the manuscript.

7. *Introduction, page 3, "interstitial tissue": It is unclear what the authors mean by interstitial tissue in the brain. Perhaps the authors mean "interstitial space" or "interstitial fluid".*

Response: This has been modified in the manuscript to "interstitial space".

8. *Introduction, page 4, "some species such as rabbit and sheep": The rat should be added to these species (Boulton et al. AmJPhysiol 1999).*

Response: This sentence refers to studies utilizing tracer recovery by cannulation to quantify lymphatic outflow, the rat study cited by the reviewer used a lymphatic ligation approach.

9. *Introduction, page 4, "extensions of the SAS that project extracranially": Has the route between the SAS and extracranial nerves been identified? Describe the evidence that the SAS projects extracranially.*

Response: Yes, many studies (including those cited for this statement Refs. 20, 22-25) have demonstrated that the SAS continues around cranial nerves through the foramina of

the skull. From Bradbury and Westrop, 1983 (Ref. 20): “major connexions between c.s.f. and deep cervical lymph is via prolongations of subarachnoid space around the olfactory nerves”. From Shen et al, 1985 (Ref. 22): “a ‘subarachnoidal-scleral-orbital outflow pathway’ provides a route for CSF drainage from the optic nerve SAS to intraorbital connective tissue.”

10. *Introduction, page 5, “the current paradigm suggests dual-outflow pathways for CSF”: It would be more accurate to describe the current status as “two or more outflow pathways”.*

Response: “Dual-outflow” indicates outflow through both blood and lymphatic routes as proposed by Pollay, 2010. We have revised this to “a dual-outflow system” instead of “dual-outflow pathways” as this was the specific term used by Pollay.

11. *Results, page 7 and elsewhere, “sacrificed”: “euthanized” would be better.*

Response: We prefer to use the term “sacrifice” as we have in previous publications.

12. *Results, page 7, “tracer appeared to localize along the paravascular spaces of arteries”: Because the exact location of the tracer was not determined, it would be better to say, “tracer accumulated around arteries” or “tracer stained the adventitial surface of arteries”*

Response: As discussed above, we have used “paravascular” to denote the presence of the 40 kDa tracer at the outer aspects of the arteries lining the subarachnoid space.

13. *Results, page 7, “arteries, such as the middle cerebral artery, over the convexities of the cortical hemispheres”: It would be better to refer to these vessels as “branches of the middle cerebral artery...”.*

Response: We have revised this to state “spreading within this space along the middle cerebral artery and its branches over the convexities of the cortical hemispheres”

14. *Results, page 8, “There was also entry of tracer into paravascular spaces around penetrating arteries”: I recommend, “Tracer was also found in the Virchow-Robin space around penetrating arteries”. As it surrounds the vessels, this is a “perivascular” space not a “paravascular” space.*

Response: This has been modified in the manuscript.

15. *Results, page 9, “surgically exposed”: Because this implies that the mouse was alive during surgery, it would be better to delete “surgically”.*

Response: This has been modified in the manuscript.

16. *Results, page 12, “The delay in the time for the signal to be apparent in the*

blood suggested that there did not appear to be rapid venous uptake of the tracer, implying that direct routes into the blood may not be active under these conditions.”: This statement makes the assumption that a direct route would be faster than via lymphatics, but no rationale or justification is given. A more accurate statement would be that a 25-minute delay was found between tracer infusion into the CSF and detection in the saphenous vein regardless of the route.

Response: A direct route to blood would be detected within the systemic circulation much faster than an indirect route through lymphatic vessels. This is the basis for the statement that the delay in time for the signal to be apparent implies that a lymphatic route is responsible for transport instead of a venous route. An example of direct blood uptake of a tracer can be seen after injection of free Evans blue into the skin of the paw as shown in Figure 1 of Proulx et al, JCI Insight, 2017 (Ref. 35). Signal is seen in the systemic blood within 15 to 30 seconds. We have already discussed the commonly-held assumption that blood uptake from the CSF would be much faster than lymphatic uptake in a previous response to the reviewer.

17. Results, page 13, “In sum, the dynamic imaging approaches indicate that active lymph transport outside the skull rather than direct blood outflow was primarily responsible”: This statement should be revised along the following lines: “Based on our dynamic imaging data, we conclude that lymph transport was the main route for tracer movement from the CSF into the bloodstream, and that outflow through the posterior facial vein had little contribution.”

Response: We agree and have revised the statement to: “Based on our dynamic imaging data, we conclude that lymph transport was the main route for tracer movement from the CSF into the systemic bloodstream rather than direct blood outflow through the posterior facial vein.”

18. Results, page 14, “Surprisingly, we could not detect immediate blood uptake of any tracer, with the patterns exhibiting delays before signal could be detected peripherally”: Why is this surprising? These tracers do not leak from the blood into most regions of the brain, so why would movement from the brain into blood be expected? Please modify accordingly.

Response: We are not examining tracer movement from brain into blood so the presence of the blood-brain barrier is not directly relevant. Of course, a blood-to-CSF barrier also exists, but pathways (such as arachnoid projections) for tracers from the CSF to blood have always been assumed to exist. Here, we show that these pathways are also not accessible for small molecular tracers.

19. Results, page 15, “the CSF outflow routes and the dynamics of transport to systemic blood were similar”: This statement should be revised along the following lines: “the CSF outflow routes through lymphatics and the dynamics of transport to systemic blood were similar”

Response: This has been modified in the manuscript.

20. Results, page 15, "Aging is closely associated with the development of several neurological disorders such as Alzheimer's disease and dementia." The link between neurodegenerative diseases and aging would be more clear if described as, "The incidence of Alzheimer's disease and other dementias and many other neurodegenerative diseases increases with age."

Response: This has been modified in the manuscript.

21. Results, page 15, "hypothesis for the development of these disorders has been developed that proposes": More straightforward: "hypothesis for the development of these disorders was proposed"

Response: This has been modified in the manuscript.

22. Results, page 16, "However, no significant difference could be seen at 60 min indicating a delayed outflow pattern in aged mice." The reasoning here is unclear. If the slope was less steep in aged mice, shouldn't the accumulation in these mice be less at 60 minutes? Please modify accordingly.

Response: The signals in the lymph node are not "accumulation" of tracer. The pegylated tracer passes through the lymph nodes without significant retention, therefore, at some point the signals will plateau and decrease. At 60 min in young mice, this has likely already occurred, while in aged mice with slower dynamics the signals may still be increasing.

23. Results, page 17, "...indicated a significant delay in lymphatic outflow from CSF in aged mice." The word "delay" is confusing. The data seem to fit better with, "...indicated significantly slower CSF outflow into lymphatics of aged mice." Please modify accordingly.

Response: This has been modified in the manuscript.

24. Discussion, page 18, "Our study is, to our knowledge, the first...": This would be a good place to cite the recent paper on fluorescence imaging of lymphatic outflow of CSF in mice by Kwon et al. (*J Immunol Methods*, 2017).

Response: We disagree. The recent report by Kwon et al. does not characterize in detail the lymphatic outflow pathways from the skull. The injections of indocyanine green in that paper were performed into the intrathecal space of the spine in high volumes (10 to 30 μ L) and lymphatic outflow was shown from the spine as well as to mandibular lymph nodes. Neither the outflow pathway to deep cervical lymph nodes was shown nor were the anatomical pathways along cranial nerves demonstrated.

25. Discussion, pages 19-20, "We found that these lymphatic vessels are

frequently discontinuous, have small diameters and, as previously reported, lack intraluminal valves (except at the base of the skull), indicating that they are different from conventional lymphatic networks.” Data from the present study should be referred to in the past tense in the Discussion.

Response: This has been modified in the manuscript.

26. Discussion, page 20, “Since the blood vessels of the dura mater tissue lack tight junctions (reference 4), it is possible that dural lymphatic vessels exist to drain this tissue specifically.”: This statement is incorrect. Reference 4 reports that some dural blood vessels have endothelial fenestrations, but does not describe the absence of tight junctions in these endothelial cells. It is unlikely that endothelial cells of any blood vessels lack tight junctions. Please modify accordingly.

Response: This has been modified in the manuscript to “Since the blood vessels of the dura mater tissue are fenestrated (Ref. 4), it is possible that the dural lymphatic vessels exist to drain this tissue.”

27. Discussion, page 20, “dural lymphatic vessels in perineurial tissue of the optic and facial nerves..”: Does this mean, “lymphatic vessels in the dura around the optic and facial nerves”? As the perineurium is one of the three sheaths around peripheral nerves, “perineurial tissue” could be misinterpreted.

Response: We did not write “perineurial” as stated above, but instead “perineural”. Nevertheless, we have modified the text as recommended.

28. Discussion, page 20, “Clearly, more research is needed to determine the importance of the dural lymphatic route for CSF outflow in comparison to the more established perineural pathways to reach extracranial lymphatic vessels.” This sentence would better fit the authors’ findings if revised as follows: “Clearly, the function of dural lymphatics deserves more research to determine whether they contribute to CSF outflow to the extent found for lymphatic vessels associated with cranial nerves and spinal nerves.”

Response: We do not see a significant difference in the meaning of these two sentences and prefer our version.

29. Discussion, page 21, “suggest that lymphatic transport is the predominant CSF outflow pathway”: “the predominant” should be changed to “an important” or “a major”. See General Comments 4-8.

Response: Please see the earlier discussion regarding the General Comments 4-8.

30. Discussion, page 21, “a lymphatic-predominant drainage of CSF”: Please revise as described in previous comment.

Response: Please see the earlier discussion regarding the General Comments 4-8.

31. Discussion, page 22, “since a continuous lining of endothelial cells with tight junctions was found to exist on the villus”: The authors’ argument against CSF movement into dural sinuses based on the presence of tight junctions should be tempered because the same argument applies to CSF movement from the SAS into lymphatics where tight junctions in the arachnoid barrier separate the two.

Response: We disagree. As shown in the ultrastructural studies (Ref. 22-25) of the SAS around exiting cranial nerves, there are disruptions or channels in the arachnoid barrier layer in these regions that allow tracers and CSF to reach the interstitial spaces. From Shen et al (Ref. 22): “The channels appeared to traverse the arachnoidal barrier layers, extending to the loose connective tissue spaces of the transitional zone between the meninges and the sclera (Fig. 3). The dimensions of the channels within the arachnoidal trabecular meshwork varied from 0.1-2.0 μm in diameter.”

32. Discussion, page 23, “To our knowledge, our study is the first to demonstrate a reduction of lymphatic outflow of CSF with aging.”: This statement should be modified to accommodate a similar statement made 10 years ago in the paper cited as reference 69, which initially described the reduction in CSF outflow with aging.

Response: We disagree. We discuss the shortcomings of reference 69 in the next sentence “One previous study has shown less outflow of radiolabeled tracers to the nasal turbinates in aged rats but did not quantify outflow to the lymphatic system or the systemic blood⁶⁹.”

33. Discussion, page 23, “An additional explanation for the reduced CSF outflow”: This would be a good place to acknowledge the age-related functional changes in the superior sagittal sinus that have been found in mice by Kang et al. (Neurobiol Aging, 2016).

Response: As we were unable to demonstrate any route to the venous blood, the reference to this work would not be relevant.

34. Conclusions, page 24, “the major exit routes”: See General comments 4-8.

Response: Please see the earlier discussion regarding the General Comments 4-8.

35. Conclusions, page 24, “there may be potential for clinical translation of the imaging technique, which could allow noninvasive monitoring of CSF outflow in patients suffering from neurological disorders”: As for the final sentence of the Abstract, the final sentence of the conclusions needs additional explanation in the Discussion to enable readers to understand what the authors are suggesting.

Response: We have now expanded on this statement in the Discussion: “In addition, there

may be potential for clinical translation of the imaging technique since noninvasive monitoring of near-infrared tracers in the blood already exists in the clinic (Ref. 72). Therefore, quantification of CSF outflow after intrathecal administration of NIR tracers in patients suffering from neurological disorders may be possible.”

36. Acknowledgements, page 35: “sharing of animal licenses”. The meaning of this statement could be misinterpreted in a negative way, where readers could think that the authors circumvented the animal licensing policy. Please clarify.

Response: We have revised this to “for cantonal-approved access to animal licenses.”